# The amyloid precursor protein is a conserved Wnt receptor

**Tengyuan Liu[1,2], Tingting Zhang[1,2], Maya Nicolas[2,3,4†], Lydie Boussicault[1], Heather Rice[3,4‡], Alessia Soldano[3,4§], Annelies Claeys[3,4], Iveta Petrova[5], Lee Fradkin[5], Bart De Strooper[3,6], Marie-Claude Potier[1], Bassem A Hassan[1]\***

[1]Paris Brain Institute – Institut du Cerveau, Sorbonne Université, Inserm, CNRS, Hôpital Pitié-Salpêtrière, Paris, France; [2]Doctoral School of Biomedical Sciences, Leuven, Belgium; [3]Center for Brain and Disease, Leuven, Belgium; [4]Center for Human Genetics, University of Leuven School of Medicine, Leuven, Belgium; [5]Laboratory of Developmental Neurobiology, Department of Molecular Cell Biology, Leiden University Medical Center, Leiden, Netherlands; [6]UK Dementia Research institute at University College London, London, United Kingdom

**\*For correspondence:**
bassem.hassan@icm-institute.org

**Present address:** [†]School of Sciences and Engineering, The American University in Cairo, Cairo, Egypt; [‡]Department of Biochemistry and Molecular Biology, Oklahoma Center for Geroscience and Heathy Brain Aging, University of Oklahoma Health Sciences Center, Oklahoma, United States; [§]Laboratory of Translational Genomics, Department of Cellular, Computational and Integrative Biology (CIBIO), University of Trento, Trento, Italy

**Competing interests:** The authors declare that no competing interests exist.

**Abstract** The Amyloid Precursor Protein (APP) and its homologues are transmembrane proteins required for various aspects of neuronal development and activity, whose molecular function is unknown. Specifically, it is unclear whether APP acts as a receptor, and if so what its ligand(s) may be. We show that APP binds the Wnt ligands Wnt3a and Wnt5a and that this binding regulates APP protein levels. Wnt3a binding promotes full-length APP (flAPP) recycling and stability. In contrast, Wnt5a promotes APP targeting to lysosomal compartments and reduces flAPP levels. A conserved Cysteine-Rich Domain (CRD) in the extracellular portion of APP is required for Wnt binding, and deletion of the CRD abrogates the effects of Wnts on flAPP levels and trafficking. Finally, loss of APP results in increased axonal and reduced dendritic growth of mouse embryonic primary cortical neurons. This phenotype can be cell-autonomously rescued by full length, but not CRD-deleted, APP and regulated by Wnt ligands in a CRD-dependent manner.

## Introduction

The Amyloid Precursor Protein (APP) is the precursor that generates the Aβ peptide, whose accumulation is associated with Alzheimer's disease (AD; *Selkoe and Hardy, 2016*). As an ancient and highly conserved protein, APP and its homologs are found across animal species in both vertebrates and invertebrates (*Shariati and De Strooper, 2013*). As a result of the alternative splicing of the 18 exons coding for APP, there are three major isoforms expressed in different organs or tissues in mice and human (*Panegyres and Atkins, 2011*). APP695 is the major isoform expressed in the brain (*Kang et al., 1987*). The expression of APP is detected at early stage during development (*Ott and Bullock, 2001*; *Sarasa et al., 2000*). In the developing mouse cortex, *App* mRNA is expressed continuously starting at embryonic day 9.5 (E9.5) coinciding with the initiation of neurogenesis and neuronal differentiation (*Salbaum and Ruddle, 1994*).

Structurally, APP is a type I transmembrane protein, which possesses a large extracellular amino acids sequence, an α−helix transmembrane sequence and a relatively short intracellular C-terminal sequence (*Müller and Zheng, 2012*; *Coburger et al., 2013*). Based on the architecture of the ectodomain, APP has been proposed to be a putative receptor (*Ninomiya et al., 1993*; *Pietrzik et al., 2004*; *Hoe et al., 2009*; *Chen et al., 2006*; *Deyts et al., 2016*). APP trafficking and processing have been intensively studied ever since the protein was first cloned. The turnover of transmembrane full-length APP is rapid (*Hunter and Brayne, 2012*; *El Ayadi et al., 2012*), and internalised APP can be degraded in lysosome or processed by α-, β-, and γ-secretase in different subcellular compartments

to produce corresponding segments of APP. (*Haass et al., 2012*; *Yuksel and Tacal, 2019*). Recently, effort has been put into researching the function of the proteolytic products of APP under normal physiological condition (*Coronel et al., 2018*), as this may provide new clues for AD research.

During *Drosophila* brain development the fly homolog of APP, called APPL, functions as key component of the neuronal Wnt-PCP signaling pathway and regulates the axonal outgrowth in fly mushroom body (*Soldano et al., 2013*). Both mammalian APP and fly APPL contain a Cysteine-Rich Domain (CRD) in the ectodomain of APP whose Cysteine distribution resembles that of the Wnt Tyrosine-protein kinase receptor Ror2 (*Coburger et al., 2013*; *Oishi et al., 2003*), suggesting the intriguing possibility that APP may itself be a receptor for Wnt family member.

Wnt signaling is an evolutionary conserved signal transduction pathway that regulates a large number of cellular processes. Three Wnt signaling pathway have been well described: the β-catenin-based canonical pathway, the planar cell polarity (PCP/Wnt) signaling pathway and the calcium pathway. Wnt signaling regulates various features during development such as cell proliferation, migration, and differentiation (*Eisenmann, 2005*). Recently, increasing evidence indicates that the Wnt signaling pathways are involved in the APP-related Aβ production (*Sellers et al., 2018*; *Elliott et al., 2018*), but the precise mode of interaction between APP and the various Wnt pathways remains unclear.

The presence of CRD in APP, the reported involvement of Wnt signaling in APP processing and the importance of Wnt signaling during development suggested to us that APP may be a novel class of Wnt receptor regulating neuronal development. We used *Drosophila* and mouse embryonic primary cortical neurons as models to explore the APP-Wnt interactions during development. We provide evidence that the CRD of APP is a conserved binding domain for both canonical and PCP Wnt ligands. Furthermore, APP trafficking and expression is regulated by Wnts through the CRD, which in turn is required for APP to regulate axonal and dendritic growth and branching.

## Results

### *Drosophila* APP like interacts genetically with Wnt5

*Drosophila* APPL has been implicated in neural development (*Cassar and Kretzschmar, 2016*; *Nicolas and Hassan, 2014*) and is required for learning and memory (*Preat and Goguel, 2016*). *Drosophila* APPL is a homologue of human APP and has been used as a model for understanding the physiological function of the APP family (*Soldano and Hassan, 2014*; *van der Kant and Goldstein, 2015*). We previously reported that *appl* genetically interacts with components of the Wnt-PCP pathway (*Soldano et al., 2013*) during mushroom body (MB) axon growth. The MBs are a bilateral neuronal structure in the fly brain required for learning and memory (*Heisenberg, 2003*). To understand the role of APPL in axonal PCP signaling, we first explored the specific nature of the genetic interaction between *appl* and the gene encoding the PCP protein Van Gough (Vang). In contrast to control MBs, 17% of male *appl* null mutant flies (*appl*[d]/Y, henceforth Appl-/-) displayed a loss of the MBb-lobe (*Figure 1A,A'*). The PCP receptor Vang is also required for β-lobe growth (*Shimizu et al., 2011*); we observed that flies homozygous for the null allele *vang*[stbm-6] exhibited 50% β-lobe loss. Whereas *vang*[stbm-6] heterozygotes show no MB defects, the loss of one copy of *vang* in Appl-/- flies is comparable (43% β-lobe loss) to the complete loss of *vang* (*Figure 1B*). Therefore, in the absence of *appl*, *vang* is haploinsufficient. Next, we performed rescue experiments of Appl-/- mutant flies. Re-expression of APPL in the mutant MBs significantly rescued β-lobe loss. In contrast, the overexpression of Vang in Appl-/- null flies failed to do so. These loss and gain of function data suggest that Wnt-PCP signaling requires APPL to regulate axonal growth (*Figure 1B*).

APPL and Vang are both transmembrane proteins that are part of the same receptor complex required for MB axonal growth (*Soldano et al., 2013*). We wondered if APPL interaction with the Wnt-PCP pathway involved a ligand and focused on *Drosophila* Wnt5 as a candidate. Wnt5 has been implicated in the regulation of MB axon growth (*van der Kant and Goldstein, 2015*; *Grillenzoni et al., 2007*), although the mechanism is unclear. We first examined the genetic interaction between *Wnt5* and *vang* in β-lobe axon growth. Loss of *vang* caused a highly penetrant phenotype (50%), while Wnt5 nulls showed β-lobe loss only in 5% of the brains examined, suggesting that Wnt5 is largely dispensable for β-lobe growth. Surprisingly, *Wnt5-/-*; *vang-/-* double mutants showed an almost complete rescue of *vang* loss of function (*Figure 1C* a,b,d,D, *Supplementary file 1*).

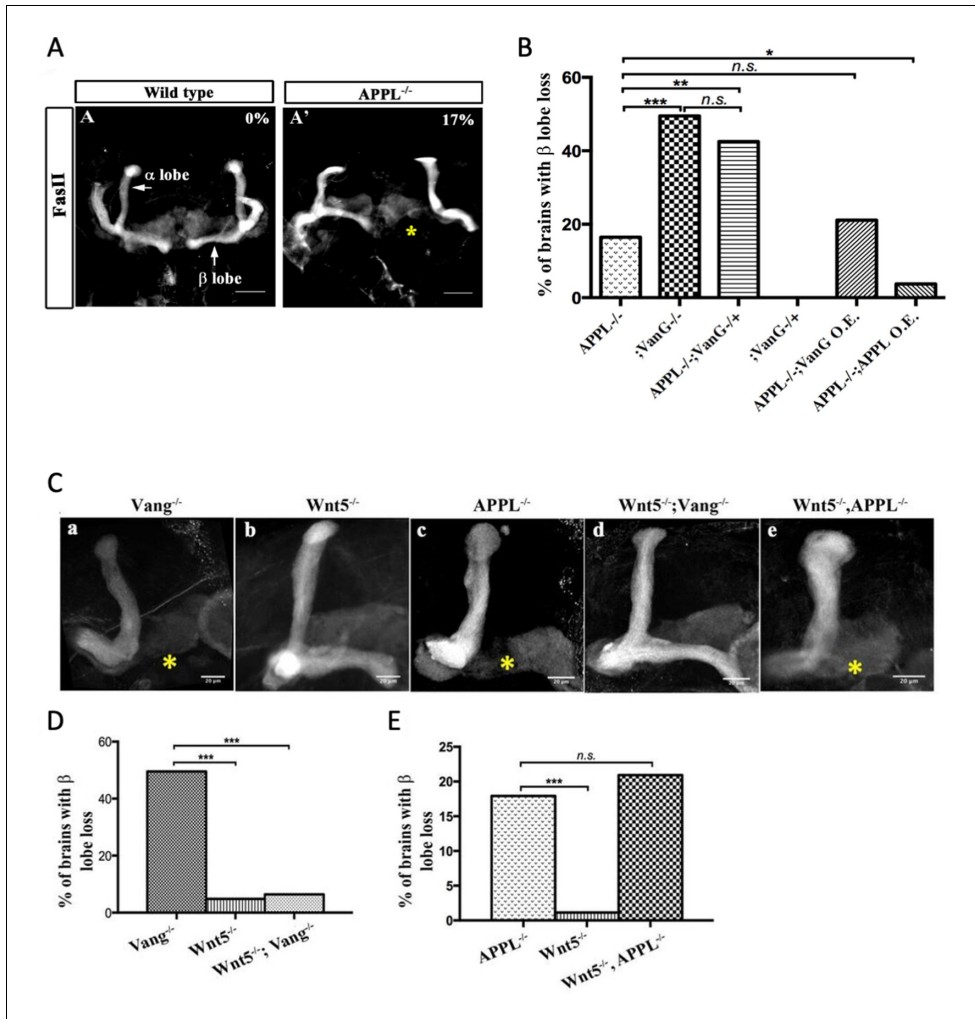

**Figure 1.** APPL mediates Wnt5a function in axonal growth. (**A–A'**) Structure of the MB neurons in adult wild type and APPL[-/-] mutant flies. Immunofluorescence using anti-FascilinII (FasII) antibody that labels the axons of the MB. (**A**) In wild type brains, the axons of the MB project dorsally to form the α lobe and medially to form the β lobe. (**A'**) In APPL null mutant flies (APPL[d]/Y referred to as APPL[-/-]), there is axonal growth defect of the β lobe (as indicated by the asterisk) in 17% of the brains examined (n=97). Images are z-projections of confocal image stacks (scale bar, 50 μm). (**B**) APPL and VanG synergistically interact and APPL is necessary for VanG activity. The histogram shows the percentage of the β lobe defect in different genetic backgrounds. The loss of Vang induced a significantly higher penetrant phenotype up to 50%, (n=103) compared to APPL[-/-]; p value = $5.18^{-7}$ calculated with G-test. The loss of one copy of Vang in wild type background had no effect on axonal growth (n=30). However, the removal of one copy of Vang in APPL[-/-] background significantly increased the phenotype to 43% (n=47) compared to APPL[-/-]; p value = 0.001026. The penetrance of the latter phenotype was not significantly different from the one observed in Vang[-/-]; p value = 0.4304. While the overexpression of APPL rescued the APPL[-/-] phenotype (4%, n=54); p value = 0.01307, the overexpression of Vang failed to (21%, n=52); p value = 0.4901. * Indicates a p value<0.05. Data are shown as median ± whiskers. (**Ca-e**) Immunofluorescence analysis using anti-FasII antibody to show the structure of the MB axons in adult mutant flies of the following genotypes: (**a**) Vang[-/-], (**b**) Wnt5[-]/Y referred to as Wnt5[-/-], (**c**) APPL[-/-], (**d**) Wnt5[-/-],Vang[-/-], and (**e**) Wnt5[-/-],APPL[-/-]. Images are z-projections of confocal image stacks (scale bar, 20 μm). The asterisks correspond to the β lobe loss phenotype. (**D**) Wnt5 inhibits axonal growth, after branching, independently of Vang. The Histogram shows the percentage of the β lobe loss phenotype. Vang[-/-] flies exhibit a highly penetrant phenotype of 50% (n=104), while Wnt5[-/-] flies show a significantly less penetrant phenotype (5%, n=103); p value = $2.33^{-14}$ calculated with G-test. The loss of Wnt5 rescued Vang loss of function (6%, n=98); p value = $4.56^{-12}$. (**E**) Wnt5 inhibits axonal growth probably through APPL. Histogram showing the penetrance of the β lobe loss phenotype. In APPL[-/-] flies, 18% of the brains tested showed an axonal defect (n=106). This percentage did not significantly change in the absence of both Wnt5 in APPL[-/-] flies (21%, n=86); p value = 0.6027. *** indicates a p value< $1^{-5}$.

Therefore, in the absence of Vang, Wnt5 inhibits β-lobe growth, suggesting that Wnt5 interacts with another receptor and antagonizes its function in PCP-mediated axon growth. We therefore examined the genetic interaction between *Wnt5* and *appl*. Loss of Wnt5 in Appl-/- flies resulted in a phenotype similar to Appl-/- flies alone (*Figure 1C* b,c,e, E). Thus, in the absence of APPL, Wnt5 no longer negatively impacts MB axon growth, suggesting that APPL may be a Wnt5 receptor.

## APPL and human APP bind Wnt5 via the CRD

Wnt5 is a member of the large family of Wnt ligands, some of whose receptors and co-receptors harbor a conserved extracellular CRD thought to be important for Wnt binding (*Dann et al., 2001*; *Oishi et al., 2003*). Intriguingly, APP harbors a CRD-like domain (*Bush et al., 1993*) in its extracellular region that includes 12 cysteine residues conserved across APP paralogs and orthologs (*Figure 2A*). The distribution of the cysteine residues resembles those present in the CRDs of other PCP receptors such as Fz and Ror-2 (*Figure 2—figure supplement 1*). We asked whether the CRDs of APP and APPL are potential Wnt5a-binding domains. To test this, we generated forms of human APP (hAPP) and APPL lacking the CRD (hAPPΔCRD and APPLΔCRD). Next, we overexpressed a tagged form of Wnt5a together with hAPP, APPL, hAPPΔCRD, or APPLΔCRD in HEK293 cells and

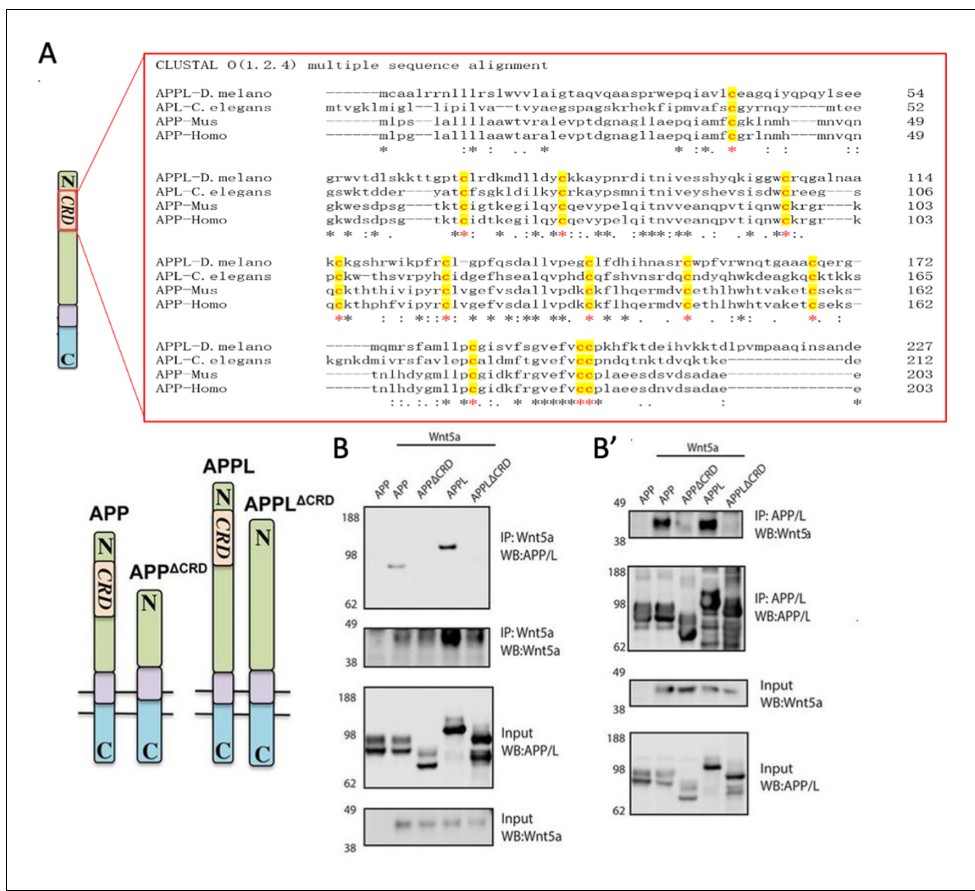

**Figure 2.** APPL and Wnt5 interact via the APP Cysteine-Rich Domain. (A) APPL extracellular region contains a conserved CRD. The figure shows a CLUSTAL alignment of the CRD of different APP homologs. The 12 cysteine residues (as indicated by the red asterisks) are highly conserved across species. (B–B') Wnt5a binds APPL and APP in a CRD-dependent manner. (B) Co-immunoprecipitation (co-IP) of the full-length proteins APP-flag and APPL-flag but not their truncated forms APPΔCRD-flag and APPLΔCRD-flag with Wnt5a-myc. (B') Reciprocal co-IP showing that Wnt5a-myc is co-IPed with flAPP-flag and APPL-flag can but not when the CRDs are deleted.

The online version of this article includes the following figure supplement(s) for figure 2:

**Figure supplement 1.** PCP receptors harbor conserved Cysteine-Rich Domains (CRD).

**Figure supplement 2.** Co-immunoprecipitation assays reveal that *Drosophila* APPL binds to WNT5.

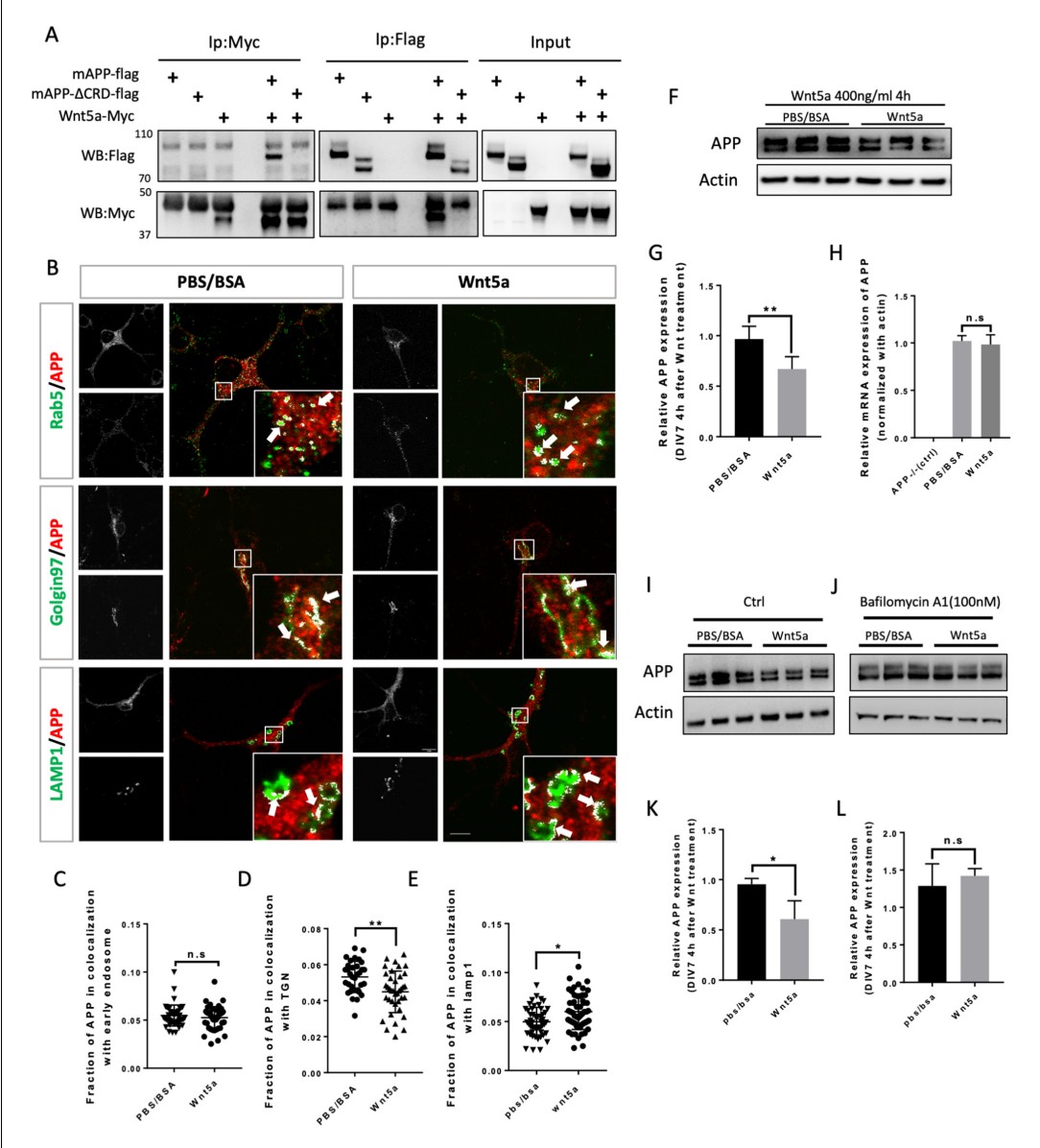

**Figure 3.** Wnt5a regulates APP expression through changing its intracellular trafficking. (**A**) Co-immunoprecipitation (co-IP) of Wnt5a-Myc with full-length proteins mAPP-flag or mAPP-delatCRD-flag. The tagged proteins were co-expressed in HEK293T cells and immunoprecipitated with ant-flag and anti-Myc antibodies. Wild-type mAPP could pull down Wnt5a and vice versa, while mAPP lacking the CRD domain showed impaired ability to pull down Wnt5a, even with higher protein levels compared to wild-type mAPP in the input. (**B**) mAPP localization after 4 hr PBS/BSA or Wnt5a treatment. Immunofluorescence for APP (red), Rab5 (early endosome marker, green), Golgin97 (TGN marker, green), and Lamp1 (lysosome marker, green) revealed mAPP localization in different intracellular compartments. The inset showed zoomed in images of the area in the white box and arrows indicated the overlap of mAPP with respective cellular compartment markers. Scale bar = 10 μm. (**C–E**) Quantification of the overlap of mAPP with early endosome (**C**), TGN (**D**) or lysosome (**E**), respectively, after Wnt5a treatment. (n=33–55 cells, t-test). (**F**) Western blots of mAPP and Actin on the lysates of DIV7 primary cultured cortical neurons showed that mAPP protein expression level was altered after Wnt5a treatment. (**G**) Quantification of the western blots results for fig F. (n=three biological independent repeat, t-test). (**H**) qPCR results showed that *mApp* mRNA expression was not affected after Wnt5a treatment on DIV7. *App-/-* mice derived primary neurons were used as a negative control. (n=three biological independent repeat, one-way ANOVA). (**I** and **J**) Western blots for APP showed that the lysosome inhibitor Bafilomycin-A (**J**) could rescue Wnt5a-induced mAPP reduction compared with control groups (**I**). (**K** and **L**) Quantification of the western blots result for figure I and J. (n=three biological independent repeat, t-test). Bars represent the mean ± S.E.M. Samples collected from at least two independent experiments. *p<0.05, **p<0.01. .

The online version of this article includes the following source data and figure supplement(s) for figure 3:

**Source data 1.** Co-immunoprecipitation (co-IP) of Wnt5a-Myc with full-length proteins mAPP-flag or mAPP-delatCRD-flag.

**Source data 2.** Western blots for mAPP and Actin after Wnt5a treatment on DIV7 primary cortical neurons.

*Figure 3 continued on next page*

*Figure 3 continued*

**Source data 3.** Western blots for mAPP and Actin after Wnt5a treatment on DIV7 primary cortical neurons.

**Source data 4.** Western blots for mAPP and Actin after adding Bafilomycin followed by Wnt5a treatment on DIV7 primary cortical neurons.

**Figure supplement 1.** Rapid turnover of fl-mAPP in culture mouse primary cortical neurons.

**Figure supplement 1—source data 1.** Western blots for time course (0.5 hr, 1 hr, 2 hr, 4 hr) of fl-mAPP expression after Cycloheximide (50 µg/ml) treatment on DIV7 primary cortical neurons.

**Figure supplement 1—source data 2.** Western blots for time course (0.5 hr, 1 hr, 2 hr, 4 hr) of fl-mAPP expression after DMSO (0.05%), treatment on DIV7 primary cortical neurons.

**Figure supplement 2.** APP overlap with early endosome, TGN and lysosome after Wnt3a/5a treatment.

**Figure supplement 3.** Wnt3a/5a treatment barely affect APP overlap with recycling endosome.

**Figure supplement 4.** Rab5 Golgin97 and Lamp1 expression after Wnt3a/5a treatment.

**Figure supplement 4—source data 1.** Western blots for fl-mAPP, Rab5, Golgin97, and Lamp1 after 4 hr of Wnt3a/5a treatment on DIV7 primary cortical neurons.

**Figure supplement 5.** Time course of fl-mAPP after Wnt3a/5a treatment.

**Figure supplement 5—source data 1.** Representative western blots for the time course (0.5 hr, 1 hr, 2 hr, 4 hr) of fl-mAPP expression after PBS/BSA (ctrl) and Wnt3a/5a treatment at DIV7.

performed co-immunoprecipitation (IP) assays. Wnt5a immunoprecipitated full-length hAPP and APPL but not hAPPΔCRD or APPLΔCRD (*Figure 2B*). Reciprocally, full-length hAPP and APPL immunoprecipitated significant amounts of Wnt5a in contrast to hAPPΔCRD and APPLΔCRD (*Figure 2B'*). Similarly, APPL was found to precipitate Wnt5 from transfected *Drosophila* S2 cell lysates (*Figure 2—figure supplement 2*).

## Wnt5a treatment affects APP trafficking and expression in maturing mouse primary cortical neuron

The findings above suggest that the APP family may represent a new class of conserved Wnt receptors. We sought to investigate this further at a cell biological level using developing mouse embryonic primary cortical neurons as a model system. APP trafficking and processing have been intensively investigated in studies relating to AD, and according to early reports the half-life of APP is quite short, ranging from 1 hr to 4 hr (*Hunter and Brayne, 2012*; *El Ayadi et al., 2012*). In mouse embryonic (E16) primary neuron cultures, full-length mouse APP (fl-mAPP; henceforth we refer to mouse APP as mAPP and to human APP as hAPP) expression significantly dropped after 2 hr of treatment with translational inhibitor (Cycloheximide) (*Figure 3—figure supplement 1*), suggesting relatively rapid turnover of mAPP. To study the relation between mAPP and Wnts, we first verified that mAPP also binds Wnt5a through its CRD and found that fl-mAPP but not mAPPΔCRD co-IP's with Wnt5a, similar to APPL and hAPP (*Figure 3A*). Next, we used immunofluorescence to localize mAPP with or without Wnt5a treatment in developing cortical neurons during axonal outgrowth (DIV7). mAPP is modified to maturation in the Trans Golgi Network (TGN) to be subsequently transferred to the plasma membrane where it can be internalized into early endosomes. From the early endosome, mAPP is either recycled back to the TGN through retromer-dependent sorting, or to the late endosome and then lysosome to be degraded (*Haass et al., 2012*; *Vagnozzi and Praticò, 2019*). We used markers for early endosomes (Rab5), recycling endosome (Rab11), TGN (Golgin97), and lysosomes (Lamp1) to trace mAPP trafficking after 2 hr and 4 hr Wnt5a treatment. The fraction of mAPP co-localized with early endosomes was not affected after 2 hr (*Figure 3—figure supplement 2A*) or 4 hr (*Figure 3B,C*) of Wnt5a treatment. Similarly, we observed no effects on co-localization with recycling endosomes (*Figure 3—figure supplement 3A,D*) indicating normal initial internalization and recycling of mAPP. However, we found a reduction of mAPP in the TGN, accompanied by an increase of mAPP in lysosomes both after 2 hr (*Figure 3—figure supplement 2D,G*) and 4 hr (*Figure 3B,D,E*) of Wnt5a treatment suggesting that Wnt5a regulates intracellular targeting of mAPP after internalization. Importantly, the levels of expression of these markers (Rab5, Golgin97, and Lamp1) are not affected by Wnt5a treatment (*Figure 3—figure supplement 4*). Next, we asked if this altered trafficking affected mAPP levels. We found that the level of fl-mAPP was significantly reduced after 4 hr of Wnt5a treatment, as shown by western blot (*Figure 3F,G*), with no effect on m*App* mRNA levels (*Figure 3H*). The results of immunofluorescence (IF) and western blot (WB) suggest that the decrease of the mAPP upon Wnt5a treatment is caused by lysosomal degradation. To

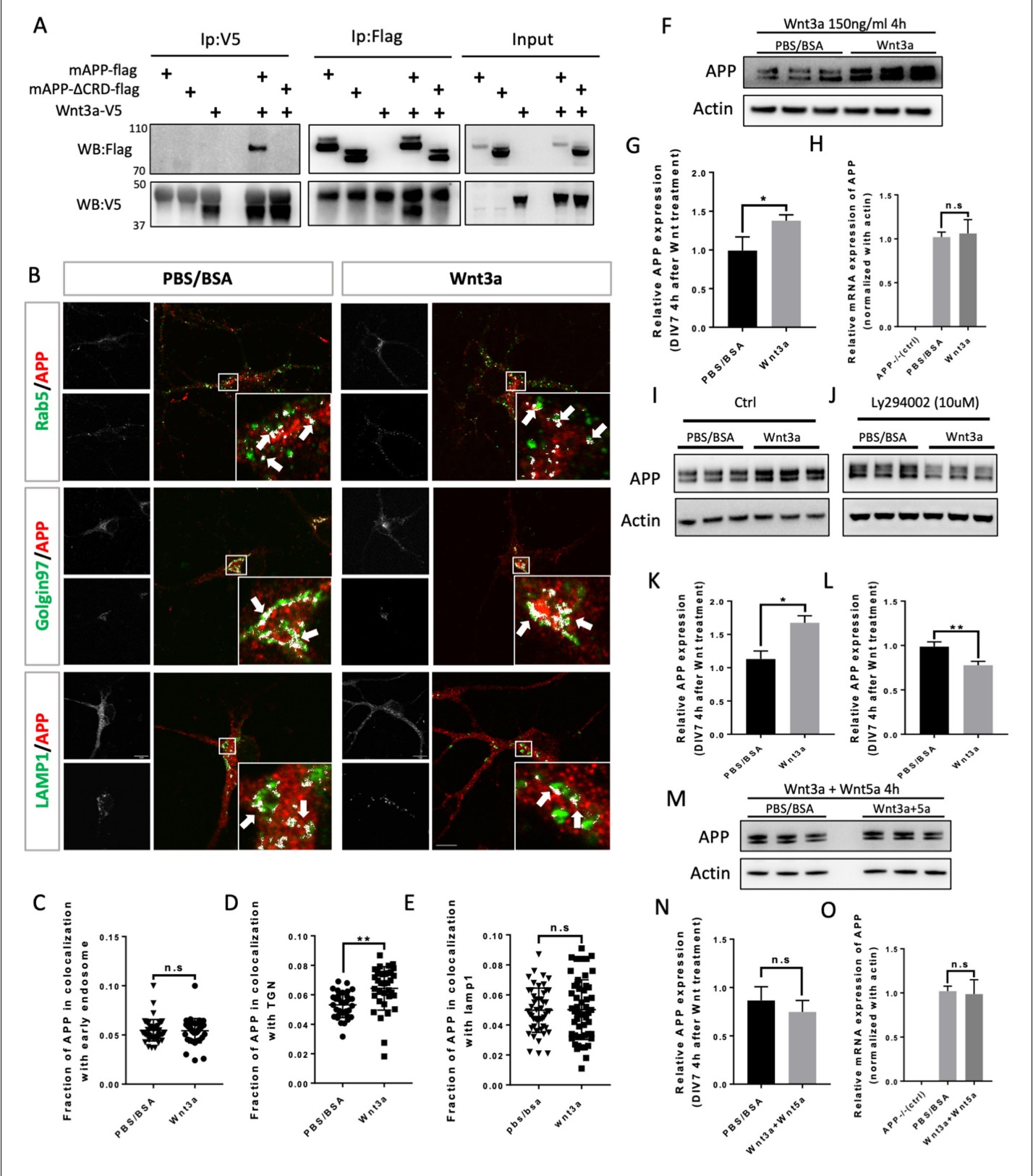

**Figure 4.** Wnt3a binds to and regulates APP expression through changing its intracellular trafficking. (**A**) Co-immunoprecipitation (co-IP) of Wnt3a-V5 with full-length proteins mAPP-flag or mAPPLΔCRD. The tagged proteins were co-expressed in HEK293T cells and immunoprecipitated with ant-flag and anti-v5 antibodies. Wild-type mAPP could pull down Wnt3a and vice versa, while mAPP lacking the CRD domain showed impaired ability to pull

*Figure 4 continued on next page*

*Figure 4 continued*

down Wnt3a even with higher protein levels compared to wild type mAPP in the input. (B) mAPP localization after 4 hr PBS/BSA or Wnt3a treatment. Immunofluorescence for APP (red), Rab5 (green), Golgin97 (green), and Lamp1 (green) revealed mAPP localization in different intracellular compartments. The inset showed zoomed in images of the area in the white box and arrows indicated the overlap of mAPP with respective cellular compartment markers. Scale bar = 10 um. (C–E) Quantification of the overlap of mAPP with early endosome (C), TGN (D) or lysosome (E), respectively, after Wnt3a treatment. (n=33–54 cells, t-test). (F) Western blots of mAPP and Actin on the lysates of DIV7 primary cultured cortical neurons showed that mAPP protein expression level was altered after Wnt3a treatment. (G) Quantification of the western blots results for figure F. (n=three biological independent repeat, t-test). (H) qPCR results showed that mApp mRNA expression was not affected after Wnt3a treatment on DIV7. App-/- mice derived primary neurons were used as a negative control. (n=three biological independent repeat, one-way ANOVA). (I and J) Western blots for APP showed that the Retromer inhibitor Ly294002 (J) could rescue Wnt3a-induced mAPP increase compared with control groups (I). (K and L) Quantification of the western blots results for figure I and J. (n=three biological independent repeat, t-test). (M) Western blots for mAPP expression 4 hr after Wnt3a and Wnt5a treatment at the same time on DIV7. PBS/BSA group acted as control group. (N) Quantification of the western blots results for figure M. (n=three biological independent repeat, t-test). (O) qPCR results for mApp mRNA expression in mApp knockout neurons (negative control), PBS/BSA treated control group neurons and Wnt3a+Wnt5a-treated group neurons. (n=three biological independent repeat, one-way ANOVA). Bars represent the mean ± S.E.M. Samples collected from at least two independent experiments. *p<0.05, **p<0.01.

The online version of this article includes the following source data and figure supplement(s) for figure 4:

**Source data 1.** Co-immunoprecipitation (co-IP) of Wnt3a-V5 with full-length proteins mAPP-flag or mAPPLΔCRD.
**Source data 2.** Western blots for mAPP and Actin after Wnt3a treatment on DIV7 primary cortical neurons.
**Source data 3.** Western blots for mAPP and Actin after Wnt3a treatment on DIV7 primary cortical neurons.
**Source data 4.** Western blots for mAPP and Actin after adding Ly294002 followed by Wnt3a treatment on DIV7 primary cortical neurons.
**Source data 5.** Western blots for mAPP and Actin after Wnt3a + Wnt5a treatment on DIV7 primary cortical neurons.
**Figure supplement 1.** APP affects beta-catenin expression after Wnt3a treatment.
**Figure supplement 1—source data 1.** Western blots for β-Catenin and Actin for APP-WT primary cultured cortical neurons after 4 hr of Wnt3a/5a treatment on DIV7 primary cortical neurons.
**Figure supplement 1—source data 2.** Western blots for β-Catenin and Actin for APP-KO primary cultured cortical neurons after 4 hr of Wnt3a/5a treatment on DIV7 primary cortical neurons.
**Figure supplement 2.** Aβ detection after Wnts treatment on DIV7 primary cortical neurons.

confirm this, we used Bafilomycin-A in combination with Wnt5a treatment. Bafilomycin-A treatment was performed 1 hr after Wnt5a addition as Wnt5a treatment already affected APP levels after 2 hr (*Figure 3—figure supplement 5*)- to inhibit the lysosome and found that this restored mAPP to control levels (*Figure 3I–L*). These data suggest that non-canonical Wnt5a-PCP signaling reduces mAPP stability.

## Wnt3a binds to and stabilizes APP via the CRD

We wondered whether mAPP can also bind other members of the Wnt family of ligands. Wnt3a is one of the 19 Wnt members in mouse and human. During development, Wnt3a usually induces β-catenin signaling pathway which plays an import role in gene expression, cell proliferation, and differentiation (*Mulligan and Cheyette, 2012*; *Rosso and Inestrosa, 2013*). Recent studies suggest that Wnt3a and beta-Catenin signaling may be involved in AD pathology (*Parr et al., 2015*; *Tapia-Rojas et al., 2016*). More interestingly, studies on mouse AD models showed that Wnt3a and Wnt5a interact competitively and antagonistically with regard to APP-mediated synapse loss (*Sellers et al., 2018*; *Elliott et al., 2018*). We therefore wondered whether, like Wnt5a, Wnt3a also binds to mAPP through the conserved CRD and regulates its levels. To test this, we performed IP experiments with Wnt3a. We found that fl-mAPP and Wnt3a co-IP in a CRD-dependent fashion (*Figure 4A*). We next tested the effects of Wnt3a treatment on APP trafficking. As shown in and *Figure 3—figure supplement 2(A)*, The fraction of mAPP co-localized with early endosomes (*Figure 4B,C*) and recycling endosome (*Figure 3—figure supplement 2A,D*) was not affected after 2 hr and 4 hr of Wnt3a treatment. However, more mAPP was present in the TGN, with no effect on the lysosomal mAPP fraction after 2 hr (*Figure 3—figure supplement 2D,G*) and 4 hr (*Figure 4B–E*) of Wnt3a treatment. The expression levels of Rab5, Golgin97 and Lamp1 were not affected after Wnt3a treatment (*Figure 3—figure supplement 4*). Western blot analysis showed increased fl-mAPP upon Wnt3a treatment (*Figure 4F,G*), but no effect on mRNA levels (*Figure 4H*). There is evidence that mAPP is recycled back to the TGN from early endosomes through the retrograde pathway (*Vagnozzi and Praticò, 2019*). To test whether Wnt3a regulates mAPP retro trafficking to the TGN, we co-treated primary neurons with Wnt3a and a retromer inhibitor (LY294002), retromer inhibitor treatment was

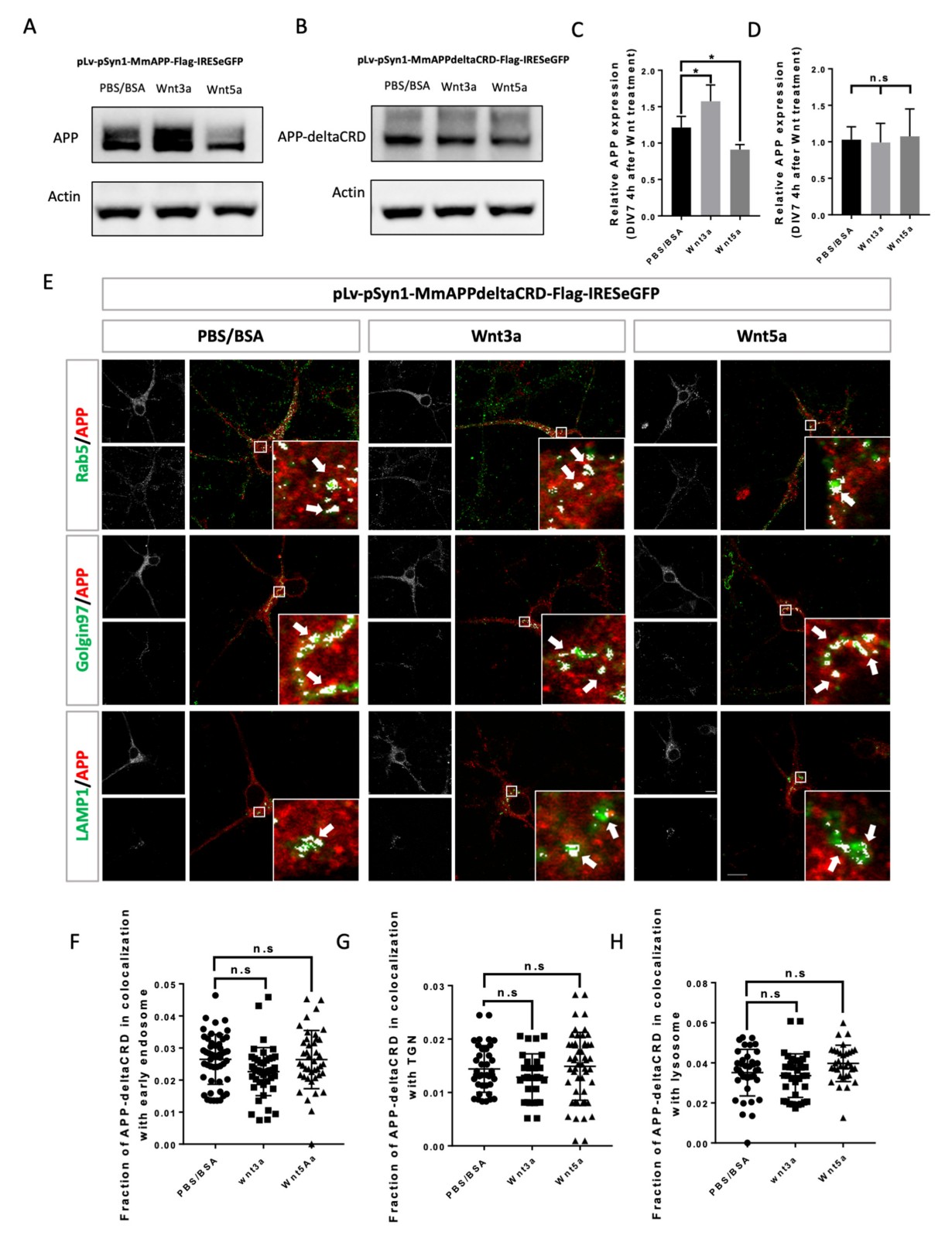

**Figure 5.** CRD is required for Wnt3a/5a to affect APP trafficking and expression. (**A and B**) Western blots for the detection of exogenous APP expression 4 hr after PBS/BSA (control), Wnt3a or Wnt5a treatment at DIV7 in APP-KO primary cultured neurons which were transfected with pLv-pSyn1-mAPP-Flag-IRESeGFP (**A**) or pLv-pSyn1-mAPPΔCRD-Flag-IRESeGFP (**B**) lenti-virus. (**C and D**) Quantification of the western blots results for figure A and B, respectively. (n=three biological independent repeat, one-way ANOVA). (**E**) Localization of exogenous mAPP expression in APP–KO primary cultured

*Figure 5 continued on next page*

*Figure 5 continued*

cortical neurons 4 hr after Wnt3a or Wnt5a treatment. Immunofluorescence for APP (red), Rab5 (green), Golgin97 (green), and Lamp1 (green) revealed mAPP localization in different intracellular compartments. The inset showed zoomed in images of the area in the white box and arrows indicated the overlap of mAPP with respective cellular compartment markers. (F–H) Quantification of the overlap of mAPP with early endosome (F), TGN (G), or lysosome (H), respectively, after Wnt3a or Wnt5a treatment. (n=32–51 cells, one-way ANOVA). Bars represent the mean ± S.E.M. Samples collected from at least two independent experiments. *p<0.05.

The online version of this article includes the following source data and figure supplement(s) for figure 5:

**Source data 1.** Western blots for APP and Actin after lenti-virus transduction followed by Wnt3a or Wnt5a treatment on DIV7 primary cortical neurons.
**Source data 2.** Western blots for APP-deltaCRD and Actin after lenti-virus transduction followed by Wnt3a or Wnt5a treatment on DIV7 primary cortical neurons.
**Figure supplement 1.** Lenti-virus-induced exogenous mAPP expressed in mAPP knock out primary cortical neuron.
**Figure supplement 1—source data 1.** Western blots for flag and Actin after lenti-virus transduction on DIV7 primary cortical neurons.
**Figure supplement 1—source data 2.** Western blots for APP and Actin after lenti-virus transduction on DIV7 primary cortical neurons.
**Figure supplement 2.** Lenti-virus induced exogenous interact with Wnts in mAPP knock out primary cortical neuron.

performed 1 hr after Wnt3a addition as Wnt3a treatment could affected APP protein expression clearly 2 hr later (*Figure 3—figure supplement 5*). This reversed the effect of Wnt3a on mAPP trafficking protein levels (*Figure 4I–L*). Then, we tested the effects of simultaneous treatment with Wnt3a and Wnt5a. This resulted in no change to APP protein levels compared to controls, suggesting that Wnt3a and Wnt5a neutralize each other's effects on mAPP (*Figure 4M,N*), again with no effects on mRNA levels (*Figure 4O*). We and others have previously shown that APP is a key component in both Wnt canonical and non-canonical Wnt signaling (*Elliott et al., 2018*; *Soldano et al., 2013*). We confirmed this in our system and found that treatment with Wnt3a resulted in β-catenin accumulation in APP-WT, but not APP-KO, primary cortical neurons at DIV7 (*Figure 4—figure supplement 1A–D*). We also observed a tendency toward a decrease in β-catenin upon Wnt5a treatment, but this did not reach statistical significance.

Because Wnt treatment affects APP trafficking and expression, we tested Aβ40 and Aβ 42 production in primary cortical neurons at DIV7 after 4 hr treatment of with Wnt3a or Wnt5a. Our data show that a 4 hr Wnt3a/5a treatment has a long-lasting effect on Aβ40/42 production and that those effects are antagonistic. Consistent with Wnt3a favoring recycling of APP, we found that Wnt3a treatment significantly decrease Aβ40/42 production (*Figure 4—figure supplement 2A,B*). In contrast, Wnt5a treatment, which induces APP internalization into acidic compartments, resulted in an increase in Aβ40/42 production (*Figure 4—figure supplement 2C,D*). Importantly, these data are in accordance with other reports that Wnt/catenin pathway favors non-amyloidogenic APP processing while Wnt/PCP signaling does the opposite (*Elliott et al., 2018*).

Taken together, these data indicate that Wnt3a also binds to mAPP via the CRD and regulates mAPP trafficking and expression and that Wnt5a and Wnt3a act antagonistically to regulate APP protein homeostasis.

## The CRD is required for Wnt-mediated regulation of APP trafficking and expression

Our data thus far show that APP interacts with Wnts through its CRD and that Wnts regulate APP intracellular trafficking and expression. We therefore asked whether the CRD is required for the effects of Wnts on mAPP. To address this question, we created two lentiviral vectors: pLv-pSyn1-mAPP-Flag-IRESeGFP (flag-tagged fl-mAPP) and pLv-pSyn1-mAPPΔCRD-Flag-IRESeGFP (flag-tagged mAPPΔCRD). Primary cortical neurons from *App* knockout mice were transduced with the fl-mAPP or mAPPΔCRD vectors, or a control GFP vector (pLv-pSyn1-IRESeGFP) exogenous APP/APPΔCRD was detectable using either anti-APP or anti-flag antibodies (*Figure 5—figure supplement 1A–C*). In neurons transduced with wild-type mAPP, we confirmed that mAPP expression could be increased and decreased by Wnt3a and Wnt5a treatments, respectively (*Figure 5A,C*). In contrast, in neurons transduced with mAPPΔCRD those effects were eliminated (*Figure 5B,D*). Finally, we performed IF to trace mAPP and mAPPΔCRD localization. Neurons transduced with wild type mAPP showed the same results as wild-type neurons with more mAPP in the TGN upon 4 hr of Wnt3a treatment and more mAPP in lysosomes upon 2 hr (*Figure 3—figure supplement 2B,E,H*) and 4 hr of Wnt5a treatment (*Figure 5—figure supplement 2A–D*). Similarly, there was no effect on

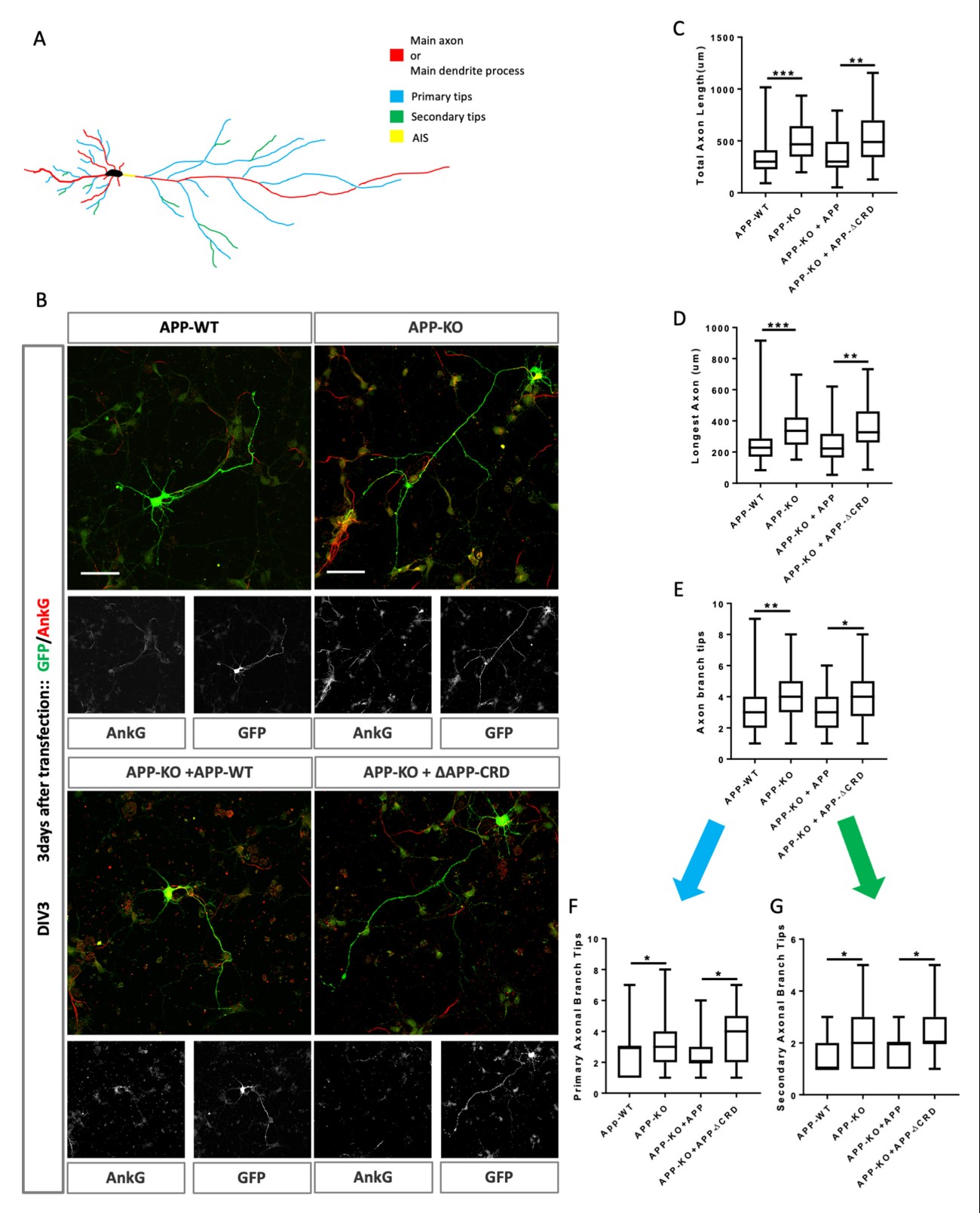

**Figure 6.** Cysteine-Rich Domain is critical for APP to regulate neurite outgrowth at DIV3. (**A**) Schematic of a primary neuron. Colored lines indicate axonal or dendritic branch tips which were quantified. Yellow indicates the Axon Initial Segment (AIS) marked with Ankry G in experiments. (**B**) Representative confocal images for GFP (green) and AnkG (red) immunostaining at DIV3 on primary cultured cortical neurons of the four genotypes examined: mAPP wild type, mAPP–KO, and mAPP–KO rescued with mAPP or CRD-mutant mAPP. Transfected plasmids containing GFP alone, mAPP-

*Figure 6 continued on next page*

*Figure 6 continued*

flag-GFP or mAPPΔCRD-flag-GFP, which was performed at the onset of cell seeding. Scale bar = 50 μm. (C–E). Quantification of three parameters on these four genotypes at DIV3: the total axon length (the main axonal process and the branches deriving from the main process, C), the longest axonal length (D), the total axonal branch tips (E), primary branch tips (F) and secondary branch tips (G). (n=59–70 cells, one-way ANOVA). Bars represent the mean ± S.E.M. Samples collected from at least two independent experiments. *p<0.05, **p<0.01 ***p<0.001.

The online version of this article includes the following figure supplement(s) for figure 6:

**Figure supplement 1.** Neurite outgrowth is unaffected in APP knock out neurons at DIV2.

**Figure supplement 2.** Analysis of dendritic outgrowth and axon complexity index at DIV3.

colocalization with recycling endosome upon 2 or 4 hr of treatment (*Figure 3—figure supplement 3, E*). All these effects were abolished in neurons transduced with mAPPΔCRD (*Figure 5E–H*; *Figure 3—figure supplement 3F*; *Figure 3—figure supplement 2C,F,I*).

In summary, these data show that the CRD of mAPP is critical for Wnt3a/5a binding and mediates the effects of Wnts on mAPP trafficking and expression.

## CRD is critical for APP to regulate neurite outgrowth and complexity

APP and its proteolytic products have been reported to affect neurite outgrowth during development (*Billnitzer et al., 2013*; *Young-Pearse et al., 2008*) in different systems. We used primary cortical neuron derived from E16.5 mice embryos to investigate if the CRD of mAPP is required for regulation of neurite outgrowth by mAPP. We examined axonal and dendritic outgrowth (*Figure 6A*) at three developmental stages in vitro: DIV2, DVI3, and DIV7 (*Dotti et al., 1988*).

While we found no effect on initial outgrowth at DIV2 (*Figure 6—figure supplement 1A–F*), at DIV3 mAPP knockout neurons exhibit increased axonal outgrowth compared to controls reflected in three parameters: total axon length, longest axon length, and the number of branch tips (*Figure 6B–E*). In contrast, dendritic outgrowth was not different from controls (*Figure 6—figure supplement 2A–C*). We asked whether the CRD was required for mAPP function during neurite outgrowth. To this end, we transfected mAPP knockout neurons with either fl-mAPP or mAPPΔCRD. Increased axonal length and axonal branch tips were rescued by the fl-mAPP but not by the form lacking the CRD at DIV3 (*Figure 6B–E*). Next, we analyzed axonal branching in greater detail and found that loss of mAPP increased the numbers of both primary and secondary axonal branches at DIV3, an increase that was rescued by fl-mAPP but not by mAPPΔCRD (*Figure 6F,G*). Finally, we examined the Axon Complexity Index (ACI) (*Wong et al., 2017*), which measures the ratio of branches of different orders to total branch number, at DIV3. At this early stage, the ACI showed a tendency to increase in mAPP knockout neurons that was not significant (*Figure 6—figure supplement 2D*), likely because both primary and secondary branches show a similar level of increase. Together these data suggest an overall increase in axonal growth. In contrast to axonal growth, we found no significant alterations in dendritic length or branching (*Figure 6—figure supplement 2E–G*) consistent with the fact that the spur in dendritic outgrowth is largely initiated at DIV4 (*Polleux and Snider, 2010*; *Barnes and Polleux, 2009*).

To further analyze neurite outgrowth, we examined axonal and dendritic growth at DIV7. By this stage, mAPP knockout neurons showed an increased ACI (*Figure 7A,B*). In contrast, total axonal length, longest axon length, and the total number of branch tips was not significantly different (*Figure 7C–E*). The increase in axonal complexity in mAPP knockout neurons was due to a significant reduction in the number of primary branches and a significant increase in the number of secondary branches (*Figure 7F,G*). Once again, all phenotypes were rescued by fl-mAPP but not mAPPΔCRD.

Finally, we examined dendritic growth at DIV7. We observed no difference in total dendrite length or the size of the longest dendrite (*Figure 7—figure supplement 1A,B*), but observed a significant decrease in the total number of dendritic processes in mAPP knockout neurons compared to controls (*Figure 7—figure supplement 1C*). This reduction was due to the presence of fewer main dendritic processes in mAPP knockout neurons but no effect was observed on the primary or secondary dendritic branches (*Figure 7—figure supplement 1D–F*). All phenotypes were rescued by fl-mAPP but not mAPPΔCRD. Taken together, our results show that the role of APP in neuronal maturation requires the CRD domain.

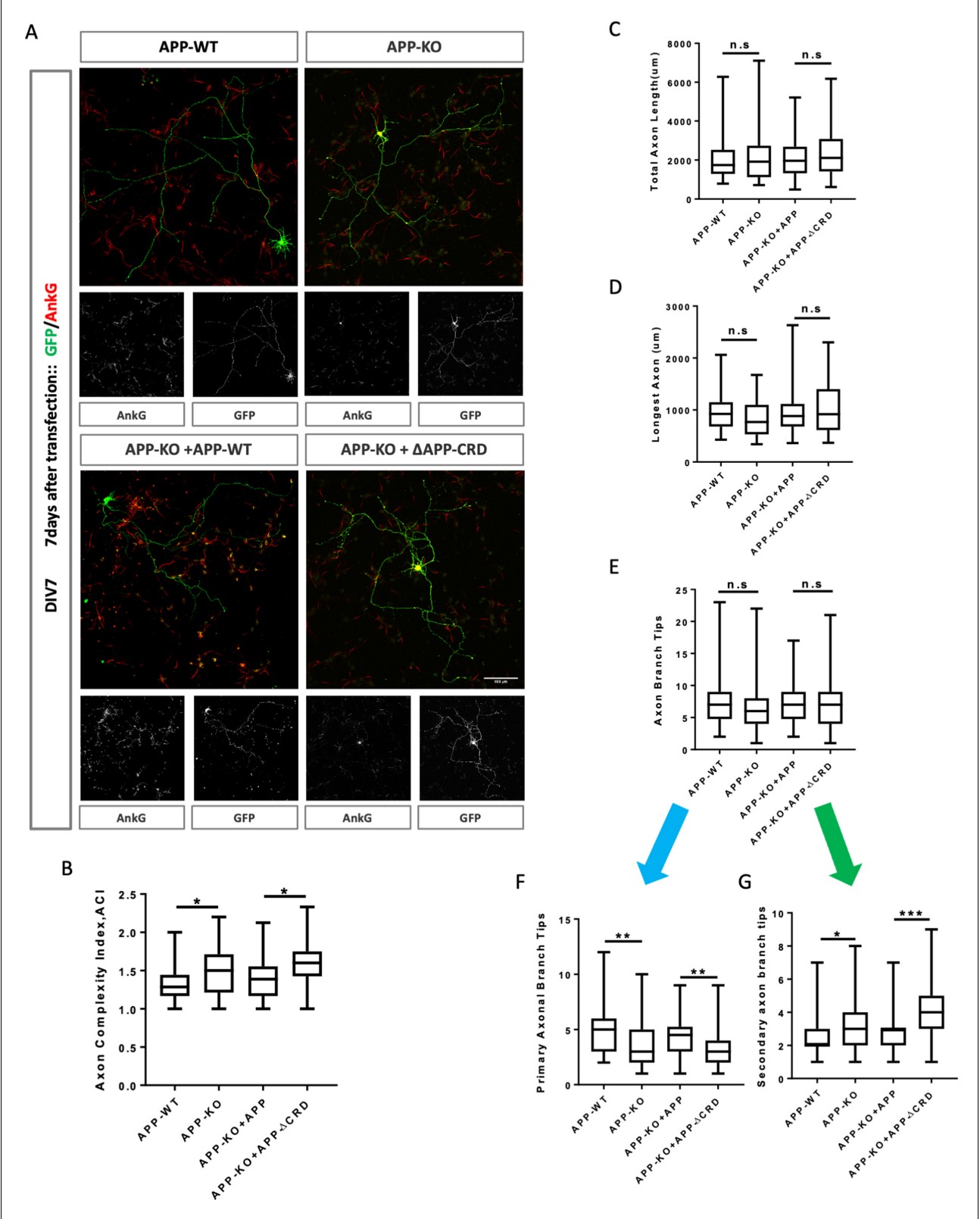

**Figure 7.** Cysteine-Rich Domain is critical for APP to regulate neurite outgrowth and complexity at DIV7. (**A**) Representative confocal images for GFP (green) and AnkG (red) immunostaining at DIV7 on primary cultured cortical neurons of the four genotypes examined: mAPP wild type, mAPP- KO and mAPP-KO rescued with APP or CRD-mutant APP. Transfected plasmids containing GFP alone, mAPP-flag-GFP or mAPPΔCRD-flag-GFP, which was performed at the onset of cell seeding. Scale bar = 100 um. (**B–G**) Analysis of Axon complexity Index (ACI, **B**), the total axonal length (**C**), the longest

*Figure 7 continued on next page*

*Figure 7 continued*

axonal length (D), all axonal branch tips (E), the primary branch tips (F) and the secondary branch tips (G) at DIV7. (n=54 cells, one-way ANOVA). Bars represent the mean ± S.E.M. Samples collected from at least two independent experiments. *p<0.05, **p<0.01, ***p<0.001.

The online version of this article includes the following figure supplement(s) for figure 7:

**Figure supplement 1.** Outgrowth and complexity analysis of neurite at DIV7.

## Wnts regulate neurite development in a CRD dependent manner

The Wnt pathway plays an important role in regulating neurite development, therefore we tested if the APP CRD is important for the interaction between APP and Wnt in regulating neurite outgrowth. To address this question, we analyzed neurite outgrowth in APP-KO primary cortical neurons rescued with APP-WT or APP-ΔCRD plasmids after Wnt3a/5a treatment. On DIV3 Wnt3a treatment significantly increased total axon length, the length of longest axon and axon branch tips in the presence of fl-APP but not APP-Δ-CRD (*Figure 8A,B,C*). These data suggest that APP mediates Wnt3a effects on axonal outgrowth. In contrast to Wnt3a, Wnt5a treatment had no effect in the presence of fl-APP but resulted in increased axonal length when the CRD domain was removed. These data show that APP normally protects axons from the effects of Wnt5a at this developmental stage and that the CRD is required for this (*Figure 8B*). Thus, APP promotes Wnt3a and antagonizes Wnt5a effects on neurons at DIV3 via the CRD domain. Similarly, on DIV7 we found a positive Wnt3a effect on axon complexity which was observed in the presence of fl-APP but not APP-Δ-CRD (*Figure 8D*). In contrast Wnt5a had no effect. Finally, we found that treatment with either Wnt3a or Wnt5a decreases the number of the main dendrites in the presence of fl-APP and the effect is absent or even reversed in the presence of APP-Δ-CRD (*Figure 8E*). Together, these data show that the CRD of APP is required for Wnt3a/5a to regulate neurite outgrowth in cortical primary neurons.

## Discussion

Here, we identify a previously unknown conserved Wnt receptor function for APP proteins. We show that APP binds both canonical and non-canonical Wnt ligands via a conserved CRD and that this binding regulates the levels of full-length APP by regulating its intracellular trafficking from early endosomes to the trans Golgi network versus the lysosome. Finally, we show that APP through the CRD regulates neurite growth and axon branching complexity in primary mouse cortical neurons.

A function for APP as a cell surface receptor has be proposed for quite a long time (*Kang et al., 1987*). The first strong evidence came from the structure of APP which shares similarity with type I transmembrane receptors. For example the growth factor like domain (GFLD) in the E1 region of APP could act as a ligand-binding site, and disulfide bridges within the same E1 area could further facilitate ligand-induced signal transduction by stabilizing the structure of APP ectodomain (*Reinhard et al., 2005*; *Rossjohn et al., 1999*; *Stahl et al., 2014*; *Kaden et al., 2009*). In addition, the site-specific proteolytic processing property of APP resembles several membrane receptors and constitute a second line of evidence. For instance, both APP and Notch are processed by γ-secretase and release the intracellular domain AICD and NICD, respectively, to induce transcriptional activity by nuclear transfer of the ICDs (*Chang et al., 2006*; *Ables et al., 2011*). Several putative ligands have been proposed including proteolytic products of APP itself. Interestingly, a conserved CRD within the E1 region of APP makes APP a putative Wnt receptor as the CRD is required for the binding between Wnt and its receptor and co-receptor like Frizzled Ror-2 and Musk (*Niehrs, 2012*), although the crystal structure of the APP CRD is different from that of classic Wnt receptors. In this study, using immunoprecipitation we have shown that mouse APP binds in a CRD-dependent manner to Wnt3a and Wnt5a the two well-known ligands for triggering canonical or non-canonical Wnt signaling pathways, respectively. Based on these data, we speculate that APP might act as a conserved Wnt receptor as its CRD is highly conserved across species, and that the binding may not be limited to Wnt3a or Wnt5a, but also apply to all other Wnt family members. As Wnt signaling is critical for development and tissue homeostasis, the role of APP as a component or putative receptor or co-receptor in Wnt signaling should be carefully studied. The difficulty to explore the exact role of APP may not only come from the complexity of Wnt signaling, but also from the structural properties of APP itself. For example, the extracellular domain of APP harbors several binding site for the

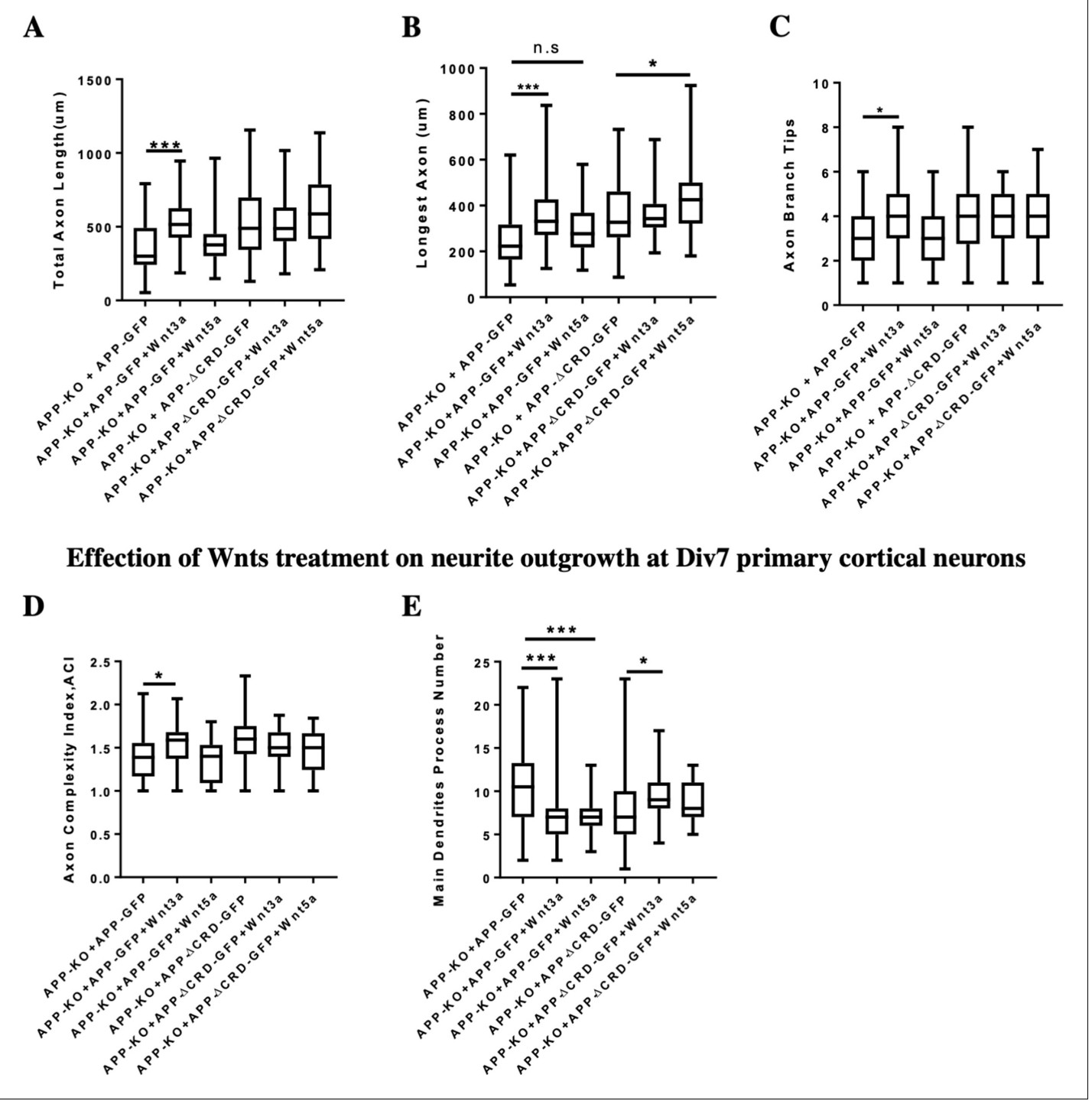

**Figure 8.** Neurite outgrowth analysis at DIV3 and DIV7 after Wnts treatment. (**A–C**) Quantification of total axon length (**A**), the longest axon length (**B**), and axon branch tips (**C**) from DIV3 cultured APP-KO primary cortical neuron after Wnts treatment rescued with APP-WT or APP-ΔCRD, respectively. (n=46–63 cells, one-way ANOVA). (**D and E**) Analysis of axon complexity (**D**) and the main dendritic process number (**E**) from DIV7 cultured APP-KO primary cortical neuron after Wnts treatment rescued with APP-WT or APP-ΔCRD, respectively. (n=47–54 cells, one-way ANOVA). Bars represent the mean ± S.E.M. Samples collected from at least two independent experiments. *P<0.05, ***p<0.001.

molecules in the extracellular matrix (ECM), such as the Heparin binding domain (HBD) is not only exist in E1 area a second HBD has been found in the E2 region (*Deyts et al., 2016*). A recent study has shown that LRP6, the co-receptor of canonical Wnt signaling, could directly bind to APP

(*Elliott et al., 2018*). Using peptide Mapping array this group further revealed several binding sites (20–70 amino acids in length) in the ectodomain of APP, and one of the sites (34 amino acids) is located within the APP CRD. Based on our IP results indicating a Wnt receptor function for APP, an obvious question arises as to whether LRP6 and APP compete for Wnt. Before trying to address this problem, several issues need to be resolved. First, as in our IP experiments the whole CRD (150 amino acids) was deleted it is not clear if specific sequences inside the CRD are critical for Wnt binding that may or may not overlap with the reported LRP6-binding site. A second important point is that the results of the peptide mapping array may not reflect the exact binging site for LRP6 as these short synthesized peptides may lack information embedded in the 3D structure like the disulfide bridges formed by Cysteines that may be critical for maintaining the stability of the APP extracellular domain (*Rossjohn et al., 1999*; *Stahl et al., 2014*). Significant further biophysical and biochemical analysis is required to understand the details of the interaction and various components of the Wnt receptor complex.

APP has been extensively reported to be involved in regulating neurite outgrowth (*Billnitzer et al., 2013*; *Young-Pearse et al., 2008*; *Araki et al., 1991*; *Southam et al., 2019*; *Small et al., 1994*; *Perez et al., 1997*), with conflicting conclusions as to whether APP promotes or inhibits neurite outgrowth. In our experiments, we found that while in *Drosophila* APPL loss reduced axonal growth, the comparison of axonal outgrowth and branching in primary cortical neuron derived from mAPP wild type or mAPP knock out mice at DIV2, DIV3 and DIV7 showed that loss of mAPP significantly accelerated axonal maturation. Specifically, we found that the initial phase of axonal growth at DIV2 is unaffected, but that APP mutant axons grow longer at DIV3 and then show increased axon complexity at DIV7. We therefore suggest that the conflicting data in the literature may arise from examining different types of neurons at different time points, where the requirement of APP may differ in a context-specific manner. We speculate that this context specificity may in part be due to the levels and types of Wnt ligands present in the environment.

Finally, our findings suggest that in addition to the well-described proteolytic processing of APP, the regulation of its recycling by Wnt ligands may be crucial for its function. It is important to note that, like proteolytic processing, Wnt ligands regulate APP stability post-translationally, as we found no effect on *App* mRNA levels upon Wnt treatment. With regard to the role of APP processing in Alzheimer's disease, recently published work suggests that an imbalance between Wnt3a/canonical signaling pathway and the Wnt5a/PCP signaling pathway at the initial step of amyloid beta production could trigger a vicious cycle favoring the amyloidogenic processing of APP (*Sellers et al., 2018*; *Elliott et al., 2018*). Our findings that Wnt3a and Wnt5a have opposite effects on amyloid beta production provide a mechanistic framework for understanding how the normal physiological function of APP directly impacts the generation of a key marker of AD, and thus potentially links the normal activity of APP to its role in neurodegeneration. Whether and how the modulation of APP's role in Wnt signaling may offer future therapeutic avenues for AD is an exciting venue for future research.

## Materials and methods

**Key resources table**

| Reagent type (species) or resource | Designation | Source or reference | Identifiers | Additional information |
|---|---|---|---|---|
| Antibody | Mouse monoclonal anti-FasII (1D4) | Developmental Studies Hybridoma Bank (DSHB) | AB_528235 RRID:AB_528235 | IF (1:50) |
| Antibody | Rabbit polyclonal anti-Wnt5a | Cell Signaling | Cat# 2392 RRID:AB_2304419 | WB (1:1000) |
| Antibody | Rabbit polyclonal anti-GFP | Invitrogen | Cat# A-11122, RRID:AB_221569 | IF (1:500) |
| Antibody | Rabbit polyclonal anti-APP | Synaptic Systems | Cat# 127 003 RRID:AB_2056967 | IF (1:100) WB (1:1000) |
| Antibody | Mouse monoclonal anti-rab5 | Synaptic Systems | Cat# 108 011 RRID:AB_887773 | IF (1:100) WB (1:1000) |

*Continued on next page*

*Continued*

| Reagent type (species) or resource | Designation | Source or reference | Identifiers | Additional information |
|---|---|---|---|---|
| Antibody | Mouse monoclonal anti-rab11a | Santa Cruz | Cat# sc-166523, RRID:AB_2173466 | IF (1:20) |
| Antibody | Mouse monoclonal anti- Golgin-97 | Invitrogen | Cat# A-21270, RRID:AB_221447 | IF (1:100) WB (1:1000) |
| Antibody | Rat monoclonal anti-Lamp1 | Santa Cruz | Cat# sc-19992, RRID:AB_2134495 | IF (1:20) WB (1:200) |
| Antibody | Chicken polyclonal anti-GFP | Abcam | Cat# ab13970, RRID:AB_300798 | IF (1:200) |
| Antibody | Guinea pig Polyclonal antiserum anti-Ankyrin G | Synaptic Systems | Cat# 386004 RRID:AB_2725774 | IF (1:100) |
| Antibody | Rabbit Polyclonal Anti-V5 | Millipore | Cat# AB3792 RRID:AB_91591 | IP (1:20) WB (1:1000) |
| Antibody | Rat Monoclonal Anti-DYKDDDDK Epitope Tag | Novus | Cat# NBP1-06712 RRID:AB_1625981 | IP (1:20) WB (1:1000) |
| Antibody | Mouse Monoclonal Anti-c-Myc | Sigma-Aldrich | Cat# M4439 RRID:AB_439694 | IP (1:20) WB (1:1000) |
| Antibody | Goat anti-Chicken IgY (H+L), Alexa Fluor488 | Invitrogen | Cat# A-11039, RRID:AB_142924 | IF (1:500) |
| Antibody | Goat anti-Rabbit IgG (H+L), Alexa Fluor488 | Invitrogen | Cat# A-11008, RRID:AB_143165 | IF (1:500) |
| Antibody | Goat anti-Rat IgG (H+L), Alexa Fluor488 | Invitrogen | Cat# A-11006, RRID:AB_141373 | IF (1:500) |
| Antibody | Goat anti- Mouse IgG (H+L), Alexa Fluor488 | Invitrogen | Cat# A-11029, RRID:AB_138404 | IF (1:500) |
| Antibody | Goat anti-Rabbit IgG (H+L), Alexa Fluor555 | Invitrogen | Cat# A-11034, RRID:AB_2576217 | IF (1:500) |
| Antibody | Goat anti-Guinea Pig IgG (H+L), Alexa Fluor555 | Invitrogen | Cat# A-21435, RRID:AB_2535856 | IF (1:500) |
| Antibody | Peroxidase AffiniPure Donkey Anti-Mouse IgG (H+L) | Jackson Immuno Research Labs | Cat# 715-035-150, RRID:AB_2340770 | WB (1:4000) |
| Antibody | Peroxidase AffiniPure Donkey Anti-Rabbit IgG (H+L) | Jackson Immuno Research Labs | Cat# 711-035-152, RRID:AB_10015282 | WB (1:4000) |
| Antibody | Peroxidase AffiniPure Donkey Anti-Rat IgG (H+L) | Jackson Immuno Research Labs | Cat# 712-035-153, RRID:AB_2340639 | WB (1:4000) |
| Chemical compound, drug | Triton X-100 | Sigma | Cat#X100 | In PBS 0.1% |
| Chemical compound, drug | Trizol Reagent | Invitrogen | Cat#15596026 | |
| Chemical compound, drug | L15 medium | Gibco | 11415064 | Medium for embryo brain isolation on ice |
| Chemical compound, drug | Mounting Medium | Vector Laboratories | Cat#H-1000 | |
| Chemical compound, drug | 0.05% trypsin/EDTA | Gibco | 25300–054 | |
| Chemical compound, drug | SVF | Invitrogen | 10270106 | |
| Chemical compound, drug | DNAse | Serlabo | LS002138 | |
| Chemical compound, drug | Neurobasal medium | Gibco | 21103049 | |
| Chemical compound, drug | B27 supplement | Gibco | 17504–044 | |

*Continued on next page*

*Continued*

| Reagent type (species) or resource | Designation | Source or reference | Identifiers | Additional information |
|---|---|---|---|---|
| Chemical compound, drug | L-glutamax | Gibco | 35050–061 | |
| Chemical compound, drug | Wnt3a | R and D Systems | 645-WN-010 | |
| Chemical compound, drug | Wnt5a | R and D Systems | 1324-WNP-010 | |
| Chemical compound, drug | Bafilomycin A1 | invitrogen | 88899-55-2 | |
| Chemical compound, drug | LY294002 | Sigma | L9908 | |
| Chemical compound, drug | Lipofectamine 3000 | Thermofisher | Cat#L3000008 | |
| Chemical compound, drug | DMEM | Gibco | Cat#10566016 | |
| Chemical compound, drug | Penicillin-Streptomycin | Gibco | Cat#15140122 | |
| Biological sample (*M. musculus*) | APP-KO mouse | A gift from Bart De Strooper's lab | N/A | |
| Cell line (*Homo-sapiens*) | Hek293 | A gift from Marie-Claude Potier's lab | N/A | |
| Sequence-based reagent | mAPP_F | This paper Ordered from IDT | PCR primers | CATCCAGAACTGGTGCAAGCG |
| Sequence-based reagent | mAPP_R | This paper Ordered from IDT | PCR primers | GACGGTGTGCCAGTGAAGATG |
| Sequence-based reagent | β-actin _F | This paper Ordered from IDT er | PCR primers | TCCATCATGAAGTGTGACGT |
| Sequence-based reagent | β-actin _R | This paper Ordered from IDT r | PCR primers | GAGCAATGATCTTGATCTTCAT |
| Transfected construct (*M. musculus*) | pLv-pSyn1-mAPP-Flag-IRESeGFP | ICM-institute, Virus facility | N/A | Lentiviral construct to transfect and express the mAPP |
| Transfected construct (*M. musculus*) | pLv-pSyn1-mAPPΔCRD-Flag-IRESeGFP | ICM-institute, Virus facility | N/A | Lentiviral construct to transfect and express the mAPP-ΔCRD |
| Transfected construct (*M. musculus*) | pLv-pSyn1- eGFP | ICM-institute, Virus facility | N/A | Lentiviral construct to transfect and express the GFP |
| Transfected construct (*M. musculus*) | pCDNA3-mApp-FLAG-IRES-eGFP | This paper | N/A | transfected construct |
| Transfected construct (*M. musculus*) | pCDNA3-mAPPΔCRD-FLAG-IRES-eGFP | This paper | N/A | transfected construct |
| Transfected construct (*M. musculus*) | pCDNA3-Wnt5a-myc | This paper | N/A | transfected construct |
| Transfected construct (*M. musculus*) | pCDNA-Wnt3A-V5 | This paper | N/A | transfected construct |
| Transfected construct (human) | pCDNA3-hApp-FLAG-IRES-eGFP | This paper | N/A | transfected construct |
| Transfected construct (human) | pCDNA3-hAPPΔCRD-FLAG-IRES-eGFP | This paper | N/A | transfected construct |
| Commercial assay or kit | QuantiTect Reverse Transcription Kit | Qiagen | Cat# 205311 | |
| Commercial assay or kit | LightCycler480 SYBR Green I Master | Roche | Cat# 04707516001 | |
| Commercial assay or kit | 4–12% polyacrylamide gels (SDS-PAGE) | ThermoFisher | NW04122BOX | |

*Continued on next page*

*Continued*

| Reagent type (species) or resource | Designation | Source or reference | Identifiers | Additional information |
|---|---|---|---|---|
| Commercial assay or kit | Protein G sepharose beads | ThermoFisher | Ref.10612D Lot.00644644 | |
| Other | Nikon | A1-R | | |
| Other | Olympus | FV-1200 | | |
| Other | DAPI | Sigma | Cat# D9564 | 1 ug/ml |
| Software, algorithm | GraphPad Prism software | GraphPad Prism (https://graphpad.com) | RRID:SCR_015807 | |
| Software, algorithm | ImageJ software | ImageJ (http://imagej.nih.gov/ij/) | RRID:SCR_003070 | |

## *Drosophila* stocks and maintenance

Flies were raised at 25°C, on standard cornmeal and molasses medium. The stocks used in this study are: w*, Appl$^d$; Vang$^{stbm-6}$; w1118, Wnt5$^{400}$; P247Gal4; w*, Appl$^d$, Wnt5$^{400}$.

## Cloning

All constructs were generated by PCR amplification and overlap extension PCR. PCR products were inserted into the respective vectors by classical restriction enzyme cloning. All constructs were sequence-verified. To generate transgenic flies, open-reading frames with epitope tags were cloned into the pUAST-attB fly expression vector and transgenes were inserted into the genome at the VK37 docking site (2L, 22A3) via PhiC31-mediated transgenesis.

## Mushroom body analyses

Adult fly brains were dissected in phosphate buffered saline (PBS) and fixed in 3.7% formaldehyde in PBT (PBS+ 0.1% Triton100-X) for 15 min. Then, the brains were washed three times in PBT and blocked in PAX-DG for 1 hr at RT. The samples were later incubated with the primary antibody overnight at 4°C. After incubation, the brains were washed three times with PBT and incubated with an ALEXA Fluor secondary antibodies (Life technologies) for 2 hr at RT. After three times washes in PBT, the brains were mounted in Vectashield (Vector Labs, USA) mounting medium. The following antibodies were used: mouse anti-FasII (Developmental Studies Hybridoma Bank (DSHB), 1/50), rabbit anti-GFP (Invitrogen, 1/500). The mounted brains were imaged on a LEICA DM 6000 CS microscope coupled to a LEICA CTR 6500 confocal system and a Nikon A1-R confocal (Nikon) mounted on a Nikon inverted microscope (Nikon). The pictures were then processed using ImageJ.

## Primary cortical neuron culture, virus transduction, and plasmids transfection

All experiments were done according to policies on the care and use of laboratory animals of European Communities Council Directive (2010/63). The protocols were approved by the French Research Ministry following evaluation by a specialized ethics committee (dossier number 4437). APP knock out mice were a gift from the De Strooper lab. Cortical primary neuron cultures were prepared from embryonic day 16.5 mice (APP wild or APP mutant), as described previously (*Cheng et al., 2016*).

Virus (pLv-pSyn1-mAPP-Flag-IRESeGFP, pLv-pSyn1-mAPPΔCRD-Flag-IRESeGFP, or pLv-pSyn1-eGFP) transduction was performed during seeding in 24-well plates (4x10$^5$cells/mL). Fifty μL (50 μL par well) of the adequate lentiviral dilution in the medium of interest must be ready in tubes. Seed 150 μL of the cell preparation to each well. Immediately add 50 μl of the diluted lentiviral preparation to each well (final MOI 2). Mix slowly the cells-lentivirus suspension by pipetting. Incubate 1 hr at 37°C. Finally add 800 μL of culture medium to each well and incubate for 3 additional days before any analysis.

Plasmids (pLv794_pTrip_PromSynaptin1_GFP_DeltaU3, pLv-pSyn1-MmApp-FLAG-IRES-eGFP, or pLv-pSyn1-mAPPΔCRD-FLAG-IRES-eGFP) transfection was performed at the onset of cell seeding (4x10$^5$cells/mL) in 24 wells plates with coverslip coated with PDL 24 hr before. All procedure follow

the protocol from Lipofectamine 3000 transfection reagent (Thermofisher Catalog Number: L3000008) with little modified, each well transfection with 500 ng corresponding plasmid, medium was refreshed 5–6 hr after transfection.

## Wnt and inhibitor treatment in primary neuron

Wnt5a (400 ng/ml) (645-WN-010, R and D Systems), Wnt3a (150 ng/ml) (1324-WNP-010, R and D Systems), and PBS/BSA (control) addition performed at DIV7. In all experiments related to inhibitor, cells will be treated with inhibitor 1 hr after Wnt addition (Bafilomycin A1, 100 nM, Invitrogen, 88899-55-2), LY294002 (10 μM, Sigma, L9908) and a DMSO (0.05%DMSO in culture medium) group will be set as control. Protein or RNA samples collected after 4 hr of Wnt treatment.

## Aβ 40/42 detection

For the Aβ 40/42 detection, first, culture medium refreshed with Wnt3a (50 ng/ml or 150 ng/ml) or Wnt5a (100 ng/ml or 400 ng/ml) on DIV7 primary cortical neurons. Then, supernatant was collected after 4 hr Wnt3a/5a treatment for Aβ detection, and refreshed with medium without Wnt3a/5a. At last, the second round supernatant collection performed after additional 24 hr culture. The procedure of Aβ detection is based on the protocol from V-PLEX Plus Aβ Peptide Panel 1 (6E10) Kit (K15200G, MSD).

## Amino acids sequence of deleted CRD

Amino acids with underscore are deleted in mouse, human APPΔCRD or APPLΔCRD.

*hAPP695* (NCBI Reference Sequence: NP_958817.1)
MLPGLALLLLAAWTARALEVPTDGNAGLLAEPQIAMF<u>CGRLNMHMNVQNGKWDSDPSGTKTC IDTKEGILQYCQEVYPELQITNVVEANQPVTIQNWCKRGRKQCKTHPHFVIPYRCLVGEFVSDAL LVPDKCKFLHQERMDVCETHLHWHTVAKETCSEKSTNLHDYGMLLPCGIDKFRGVEFVCC</u>PLA EESDNVDSADAEEDDSDVWWGGADTDYADGSEDKVVEVAEEEEVAEVEEEEADDDEDDEDGD EVEEEAAEEPYEEATERTTSIATTTTTTTTESVEEVVRVPTTAASTPDAVDKYLETPGDENEHAHFQ KAKERLEAKHRERMSQVMREWEEAERQAKNLPKADKKAVIQHFQEKVESLEQEAANERQQL VETHMARVEAMLNDRRRLALENYITALQAVPPRPRHVFNMLKKYVRAEQKDRQHTLKHFEHV RMVDPKKAAQIRSQVMTHLRVIYERMNQSLSLLYNVPAVAEEIQDEVDELLQKEQNYSDDVLAN MISEPRISYGNDALMPSLTETKTTVELLPVNGEFSLDDLQPWHSFGADSVPANTENEVEPVDAR PAADRGLTTRPGSGLTNIKTEEISEVKMDAEFRHDSGYEVHHQKLVFFAEDVGSNKGAIIGLMVG GVVIATVIVITLVMLKKKQYTSIHHGVVEVDAAVTPEERHLSKMQQNGYENPTYKFFEQMQN.

*MmApp* (NCBI Reference Sequence: NP_031497.2)
MLPSLALLLLAAWTVRALEVPTDGNAGLLAEPQIAMF<u>CGKLNMHMNVQNGKWESDPSGTKTC IGTKEGILQYCQEVYPELQITNVVEANQPVTIQNWCKRGRKQCKTHTHIVIPYRCLVGEFVSDALL VPDKCKFLHQERMDVCETHLHWHTVAKETCSEKSTNLHDYGMLLPCGIDKFRGVEFVCC</u>PLAE ESDSVDSADAEEDDSDVWWGGADTDYADGGEDKVVEVAEEEEVADVEEEEADDDEDVEDGDEV EEEAEEPYEEATERTTSTATTTTTTTESVEEVVRVPTTAASTPDAVDKYLETPGDENEHAHFQKAK ERLEAKHRERMSQVMREWEEAERQAKNLPKADKKAVIQHFQEKVESLEQEAANERQQLVETHM ARVEAMLNDRRRLALENYITALQAVPPRPHHVFNMLKKYVRAEQKDRQHTLKHFEHVRMVDPKK AAQIRSQVMTHLRVIYERMNQSLSLLYNVPAVAEEIQDEVDELLQKEQNYSDDVLANMISEPRISYG NDALMPSLTETKTTVELLPVNGEFSLDDLQPWHPFGVDSVPANTENEVEPVDARPAADRGLTT RPGSGLTNIKTEEISEVKMDAEFGHDSGFEVRHQKLVFFAEDVGSNKGAIIGLMVGGVVIATVIVITL VMLKKKQYTSIHHGVVEVDAAVTPEERHLSKMQQNGYENPTYKFFEQMQN.

*appl* (*Drosophila melanogaster*) (NCBI Reference Sequence: NP_001245451.1)
mcaalrrnlllrslwvvlaigtaqvqaasprwepqiavl<u>ceagqiyqpqylseegrwvtdlskkttgptclrdkmdlldyc kkaypnrditnivesshyqkiggwcrqgalnaakckgshrwikpfrclgpfqsdallvpegclfdhihnasrcwpfvrwn qtgaaacqergmqmrsfamllpcgisvfsgvefvcc</u>pkhfktdeihvkktdlpvmpaaqinsandelvmndeddsn dsnyskdaneddlddeddlmgddeeddmvadeaataggspntgssgdsnsgslddinaeydsgeegdnyeedg agseseaeveaswdqsggakvvslksdsssppsapvapapekapvksesvtstpqlsasaaafvaansgnsgtgaga

ppstaqptsdpyfthfdphyehqsykrleeshrekvtrvmkdwsdleekyqdmrladpkaaqsfkqrmtarfqtsv
qaleeegnaekhqlaamhqqrvlahinqrkreamtcytqalteqppnahhvekclqkllralhkdrahalahyrhlln
sggpggleaaaserprtlerlididravnqsmtmlkrypelsakiaqlmndyilalrskddipgsslgmseeaeagildk
yrveierkvaekerlrlaekqrkeqraaereklreeklrleakkvddmlksqvaeqqsqptqsstqsqaqqqqqekslp
gkelgpdaalvtaanpnlettksekdlsdteygeatvsstkvqtvlptvdddavqravedvaaavahqeaepqvqhf
mthdlghressfslrrefaqhahaakegrnvyftlsfagialmaavfvgvavakwrtsrsphaqgfievdqnvtthhpiv
reekivpnmqingyenptykyfevke.

## Quantitative real-time PCR (qRT-PCR)

Cells were lysed for RNA or protein extraction and then subjected to qRT-PCR or western blots as previously described (*Liu et al., 2014*). The detailed sequence of each primer used in the whole study for qRT-PCR were summarized below : β-actin, sense 5'- TCCATCATGAAGTGTGACGT-3' and anti-sense 5'- GAGCAATGATCTTGATCTTCAT −3', mAPP, sense 5'- CATCCAGAACTGG TGCAAGCG-3' and anti-sense 5'- GACGGTGTGCCAGTGAAGATG −3' GAPDH, sense 5'- GC TGCCAAGGCTGTGGGCAAG-3' and anti-sense 5'- GCCTGCTTCACCACCTTC −3'.

## Western blots

Western blots were performed follow the user guide of Mini Gel Tank (ThermoFisher, A25977) with little modified. Briefly, Protein samples collected from total cell lysates with RIPA buffer, supernatant were collected after centrifugation, denatured samples were loaded separated on the 4–12% poly-acrylamide gels (SDS-PAGE) (ThermoFisher, NW04122BOX) and then transferred to the 0.42 μm nitrocellulose membranes, blots visualization performed after primary and secondary antibody incubation.

## Immunoprecipitation

Human embryonic kidney (HEK293) cells (provided from Dr. Marie-Claude Potier) were purchased from ATCC, regularly mycoplasma test performed by the CELIS core facility of ICM. For the immuno-precipitation experiments, HEK293in 10 cm dish (70% confluent) were transfected with pCDNA3-MmApp-FLAG-IRES-eGFP, pCDNA3-mAPPΔCRD-FLAG-IRES-eGFP, pCDNA3-Wnt5a-myc, pCDNA-Wnt3A-V5 or co-transfected APP or APPΔCRD with Wnt3a or Wnt5a. Three days after transfection, cells were collected with NP-40 lysis buffer, then sample supernatant was collected after >12000 rpm centrifugation for 20 min at 4°C, 450 μl supernatant was incubated with primary antibody over-night at 4°C, then Protein G sepharose beads (Thermo Fisher Scientific) were added to the sample to capture protein-antibody complex by rotating 2 hr at room temperature, then washed four times with the lysis buffer, and resuspended with loading buffer then denatured at 95 degree for 10 min, blots visualized after western blot procedure as described before.

## Immunofluorescence

At DIV 7, cultured primary neurons in 24 wells were washed once with 1X PBS, then fixed in 4% para-formaldehyde (PFA) in PBS at room temperature (RT) for 10 min. After three times washing with 1X PBS, cells were blocked with 10% normal donkey or goat serum in 1 X PBS for 30 min at RT followed by three times washing in 1 X PBS. Thereafter, cells were incubated with primary antibodies diluted in 1 X PBS containing 1% normal donkey or goat serum for 2–3 hr at RT. three times washing with 1 X PBS, incubated with appropriate secondary antibodies conjugated with Alexa Fluor 488, Alexa Fluor 555, or Alexa Fluor 647 (1:500, Invitrogen) in 1 X PBS containing 1% normal donkey or goat serum for 1 hr at RT. Washed with 1 X PBS for three times, then counterstained the slides with DAPI (1:2000, Sigma) and mounted by using Vectashield (Vector) after rinsing. Primary antibodies used in this study were rabbit anti-APP (1:100, Synaptic Systems, 127 003), mouse anti-rab5 (1:100, Synaptic Systems, 108011), mouse anti-rab11a (1:20, Santa Cruz, sc-166523), mouse anti- Golgin-97 (1:100, Invitrogen, A-21270), rat anti-Lamp1 (1:20, Santa Cruz, sc-19992). After staining, images were obtained by using confocal microscope (Olympus FV-1200 or Leica SP8). The percentage of APP or APP-ΔCRD co-localizing with rab5, rab11, Golgin-97 and Lamp1 was calculated using JACOP (*Bolte and Cordelières, 2006*).

## Statistical analyses

Statistical analyses were performed using GraphPad Prism software (GraphPad Software Inc, La Jolla, CA, USA). Differences between groups were compared using the Student's t test, One-way ANOVA and Mann Whitney two-sample test (two-tail) as appropriate.

## Acknowledgements

We thank Dr. Radoslaw Ejsmont for writing the co-localization macro, Natalia Danda for the construction of the plasmids used in this work, Drs. Ariane Ramaekers, Natalia Mora Garcia, Gerit Linneweber, Simon Weinberger and Guangda Liu for helpful discussions. We thank Drs. Zeynep Kalender Atak and Marina Naval Sanchez for support on the statistical analysis of the data and Dr. Jean-Maurice Dura for fly lines. Mouse breeding work was conducted at the PHENO-ICMice facility. The Core is supported by 2 Investissements d'Avenir grants (ANR-10- IAIHU-06 and ANR-11-INBS-0011-NeurATRIS) and the 'Fondation pour la Recherche Médicale'. Primary neuron culture work was carried out at the CELIS core facility with support from Program Investissements d'Avenir (ANR-10-IAIHU-06). Light microscopy was carried out at the ICM.Quant facility. We thank all core technical staff involved. We thank Philippe Ravassard for providing the pSYN lentiviral vector backbone and Blandine Bonnamy and Clementine Ripoll from iVECTOR core facolity for technical assistance and production of the lentiviral vectors. This work was supported by Fonds Wetenschappelijke Onderzoeks (FWO) grants G.0543.08, G.0680.10, G.0681.10 and G.0503.12, the program 'Investissements d'avenir' ANR-10-IAIHU-06, the Einstein-BIH program, the Neuro-Glia Foundation, a Sorbonne Université Emergence grant, an Allen Distinguished Investigator Award and the Roger De Spoelberch Foundation Prize (to BAH), ANR-12-MALZ-0004 grant (to MCP), the Vlaams Instituut voor Biotechnologie and the Methusalem grants from KU Leuven (to BDS and BAH), the Nederlandse Organisatie voor Wetenschappelijk Onderzoek (NWO; ZonMw TOP grant 40-00812-98-10058) and the Hersenstichting Nederland [HS 2011(1)−46] (to LGF), a doctoral fellowship from the Centre National de Recherche Scientifique Libanais (to MN) and Chinese Scholarship Council fellowships (to TZ and TL). The authors declare no competing financial interests.

## Additional information

### Funding

| Funder | Grant reference number | Author |
|---|---|---|
| China Scholarship Council | | Tengyuan Liu |
| Hersenstichting | HS 2011(1)-46 | Lee Fradkin |
| Lebanese National Council for Scientific Research | | Maya Nicolas |
| Vlaams Instituut voor Biotechnologie | | Bart De Strooper Bassem A Hassan |
| Fondation Roger de Spoelberch | 1911001IN | Bassem A Hassan |
| Agence Nationale de la Recherche | ANR-10-IAIHU-06 | Marie-Claude Potier Bassem A Hassan |
| Fonds Wetenschappelijk Onderzoek | G.0543.08 | Bassem A Hassan |
| Neuron-Glia Foundation | 2003009NA | Bassem A Hassan |
| KU Leuven | Methusalem | Bart De Strooper Bassem A Hassan |
| Nederlandse Organisatie voor Wetenschappelijk Onderzoek | 40-00812-98-10058 | Lee Fradkin |
| Fonds Wetenschappelijk Onderzoek | G.0680.10 | Bassem A Hassan |
| Fonds Wetenschappelijk On- | G.0681.10 | Bassem A Hassan |

| | | |
|---|---|---|
| derzoek | | |
| Fonds Wetenschappelijk On-derzoek | G.0503.12 | Bassem A Hassan |
| Agence Nationale de la Re-cherche | ANR-11-INBS-0011-NeurATRIS | Marie-Claude Potier Bassem A Hassan |
| Agence Nationale de la Re-cherche | ANR-12-MALZ-0004 | Marie-Claude Potier Bassem A Hassan |

The funders had no role in study design, data collection and interpretation, or the decision to submit the work for publication.

## Author contributions
Tengyuan Liu, Conceptualization, Formal analysis, Investigation, Methodology, Writing - original draft, Writing - review and editing; Tingting Zhang, Formal analysis, Investigation, Methodology; Maya Nicolas, Conceptualization, Formal analysis, Investigation, Methodology, Writing - original draft; Lydie Boussicault, Heather Rice, Investigation, Methodology; Alessia Soldano, Conceptualization, Investigation; Annelies Claeys, Iveta Petrova, Investigation; Lee Fradkin, Conceptualization, Supervision, Funding acquisition; Bart De Strooper, Marie-Claude Potier, Supervision, Funding acquisition; Bassem A Hassan, Conceptualization, Supervision, Funding acquisition, Writing - original draft, Project administration, Writing - review and editing

## Author ORCIDs
Maya Nicolas https://orcid.org/0000-0002-7148-6357
Alessia Soldano http://orcid.org/0000-0002-3120-9929
Bart De Strooper http://orcid.org/0000-0001-5455-5819
Marie-Claude Potier http://orcid.org/0000-0003-2462-7150
Bassem A Hassan https://orcid.org/0000-0001-9533-4908

## Ethics
Animal experimentation: All experiments were done according to policies on the care and use of laboratory animals of European Communities Council Directive (2010/63). The protocols were approved by the French Research Ministry following evaluation by a specialized ethics committee (dossier number 4437).

## Decision letter and Author response
Decision letter https://doi.org/10.7554/eLife.69199.sa1
Author response https://doi.org/10.7554/eLife.69199.sa2

# Additional files
## Supplementary files
• Supplementary file 1. Genetic interaction between Wnt5 and Vang. The table lists the penetrance of the phenotype and the number of brains analyzed in the Vang-Wnt5 genetic interaction experiment.

• Transparent reporting form

## Data availability
No new datasets were generated in this study.

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
