## [Decision Letter]

[Editors' note: this paper was reviewed by Review Commons.]

**Acceptance summary:**

Your manuscript documents interactions between APP and Wnt ligands. Interestingly, both the canonical Wnt ligand Wnt3a and noncanonical Wnt ligand Wnt5a are able to bind to a conserved Cysteine Rich Domain (CRD) in the extracellular portion of APP, Your work provides compelling evidence that the CRD is critical for APP to regulate proper neurite outgrowth and neurite complexity. Moreover, your data indicate that Wnt3a binds to APP and promotes APP recycling and stability. In contrast, Wnt5a binds to APP and negatively affects APP stability. Overall, this study is novel and will generate interest in the fly and AD community..

---

## [Author Response]

Reviewer #1Evidence, reproducibility and clarity:In this manuscript, Liu et al. studied the interactions between APP and Wnt ligands. It was found that both canonical Wnt ligand Wnt3a and noncanonical Wnt ligand Wnt5a are able to bind to the conserved Cysteine Rich Domain (CRD) in the extracellular portion of APP, and that the CRD is critical for APP to regulate neurite outgrowth and complexity. While Wnt3a binds to APP and promotes APP recycling and stability, Wnt5a binds to APP and exhibits an opposite effect on APP stability. Overall, the study is novel and interesting.

We thank the reviewer for the positive comments on the novelty and interest of the work.

Major comments:While the findings and conclusions in this manuscript are solid, the authors should perform additional experiments to characterize the interactions between Wnt ligands and APP and determine subsequent effects on Wnt signaling and Aβ production.

We thank the reviewer for the positive comments on the strength of the evidence and address their specific comets below.

1. It is unclear whether Wnt3a and Wnt5a have the same binding site or different binding sites within the CRD of APP, and whether Wnt3a and Wnt5a compete with one another for binding.

We note that our data suggest that Wnt3a and Wnt5a do compete for APP binding. This is because while Wnt3a increases APP levels and Wnt5a decreases it, treatment with both ligands at the same time restores APP protein back to control levels. Furthermore, CRD domain folding is thought to require disulfide bonds between the Cysteine residues, and it is therefore quite unlikely that further dissection of the CRD will yield meaningful results Generally, while we agree that determining the precise biochemical properties required for APP-Wnt interactions is interesting, we believe that going into such level of detail is beyond the scope of this work.

2. It is unclear whether the interactions between Wnt ligands and APP modulate Wnt signaling. Specifically, are the canonical and noncanonical Wnt pathways activated after Wnt3a binding and Wnt5a binding, respectively?

We and others have previously shown that APP is a key component in both Wnt canonical and non-canonical signaling groups [1,3] and these findings were the reason why we tested if APP might directly interact with Wnt ligands. We have now performed IF and WB with β-catenin, Dvll and Dvl2 antibodies. Unfortunately, only the Β-catenin antibody worked in WB experiments on primary neurons. We find that Wnt3a treatment increases levels of β-Catenin in control but not APP-KO primary neurons, consistent with previous findings in cell lines that APP enhances canonical Wnt signaling. These data are now presented in Figure 4—figure supplement 1 and on page 8 of the revised manuscript.

3. It has been demonstrated that canonical Wnt signaling inhibits neurite outgrowth. In Figure 6, it was shown that mAPP knockout neurons exhibited increased axonal outgrowth, which was rescued by the fl-mAPP but not by the form lacking the CRD at DIV3. Therefore, the authors should determine whether the APP effect on neurite outgrowth is associated with an activation of canonical Wnt signaling. Specifically, does the fl-mAPP facilitate activation of canonical Wnt signaling in neurons? If yes, does the APP lacking the CRD lost its ability on canonical Wnt signaling?

This is an excellent suggestion to disentangle the effects of each of the two pathways on APP-mediated neurite outgrowth. To address this issue we treated neurons expressing either fl-APP or APP-Δ-CRD with Wnt3a or Wnt5a and measured the effect on APP-dependent axon and dendrite growth on DIV3 and DIV7.

On DIV 3 we found that at DIV3 Wnt3a treatment significantly significant increased total axon length, the length of longest axon and axon branch tips in the presence of fl-APP but not APP-Δ-CRD (revision Figure 2 A, B, C). These data suggest that APP mediates Wnt3a effects on axonal outgrowth. In contrast to Wnt3a, Wnt5a treatment had no effect in the presence of fl-APP but resulted in increased axonal length when the CRD domain was removed. These data show that APP normally protects axons from the effects of Wnt5a at this developmental stage and that the CRD is also required for this. Thus, APP promotes Wnt3a and antagonizes Wnt5a effects on neurons at DIV 3 via the CRD domain.

Similarly, on DIV7 we found a positive Wnt3a effect on axon complexity which was observed in the presence of fl-APP but not APP-Δ-CRD. In contrast Wnt5a had no effect. Finally, we found that treatment with either Wnt3a or Wnt5a decreases the number of the main dendrites in the presence of fl-APP and the effect is absent or even reversed in the presence of APP-Δ-CRD.

Together, these data show that the CRD of APP is required for Wnt3a/5a to regulate neurite outgrowth in cortical primary neurons. We will add these data to the revised manuscript.

These data are now presented in Figure 8 and on pages 11-12 of the revised manuscript.

4. It is unclear whether the interactions between Wnt ligands and APP modulate Aβ production, and whether Wnt3a and Wnt5a have different effects on Aβ production after their binding to the CRD of APP.

We have tested Aβ40 and Aβ 42 production in primary cortical neurons after 4 hours treatment of Wnt3a or Wnt5a at DIV7. Briefly, after a 4-hour treatment with Wnt3a/5a, supernatant was collected at 4 hours and at 24hours for Aβ detection using MSD kit. Our data shows that a 4-hour Wnt3a/5a treatment has a long-lasting effect on Aβ40/42 production and that those effects are antagonistic. Consistent with Wnt3a favoring recycling of APP we found that Wnt3a treatment significantly decrease Aβ40/42 production. In contrast, Wnt5a treatment, which indices APP internalization into acidic compartments, resulted in an increase in Aβ40/42 producing. Importantly, these data are in accordance with other reports that Wnt/catenin pathway favors non-amyloidogenic APP processing while Wnt/PCP signaling does the opposite.

These data are now presented in Figure 4—figure supplement 2 and on pages 8-9 of the revised manuscript.

Minor comments:1. The authors should provide detail information on the APP plasmid construct hAPPΔCRD. Specifically, which amino acid residues are removed from the extracellular portion of APP?

The exact amino acid sequence of APP/APP-ΔCRD within plasmids or virus we used in our experiments have been added to the revised manuscript in the Materials and methods section pages 18-20.

2. It has been reported that APP co-activates both the canonical and noncanonical Wnt pathways through physical interactions with Wnt co-receptors LRP6 and Vangl2, respectively (ref. 24). Interestingly, some LRP6-binding sites are also located in the CRD of APP. Therefore, more discussions are needed to address this relevance.

A discussion of this interesting point was added to the revised manuscript on pages 13-14.

Significance:APP is central to the study of AD. It is also established that dysregulation of Wnt signaling contributes to AD pathogenesis. Therefore, the study on the interactions between APP and Wnt ligands is significant, and the finding of Wnt ligand-binding to APP described in this manuscript is novel and interesting. However, in Figure 3F and Figure 4F, the concentrations of Wnt5a and Wnt3a were 400 ng/ml and 150 ng/ml, respectively, for modulating APP level/stability in DIV7 primary cortical neurons. These are quite high concentrations and difficult to be reached at pathophysiological conditions (particularly for Wnt5a). Therefore, the pathophysiological significance of the interactions between Wnt ligands and APP is unclear.

We thank the reviewer for the positive comments. We note that the concentrations we used mirror (and are even less than in some cases) what is used in the literature which is why we used them. However, we have now tested different doses in the Aβ40/42 experiments and find similar effects for Wnt3a and the same trend for lower doses of Wnt5a.

These data are now presented in Figure 4—figure supplement 2 and on pages 8-9 of the revised manuscript.

Reviewer #2Evidence, reproducibility and clarity:The manuscript submitted by Liu and colleagues described an interesting observation that the amyloid precursor protein (APP) binds to Wnt ligands Wnt3a and Wnt5a; this binding regulates APP protein level and neuronal development in vivo and in vitro. Authors used multiple approaches to show that APP is a Wnt receptor for Wnt5a and Wnt3a, including the use of a *Drosophila* genetic model, APP KO, and a neuronal lentivirus-mediated transduction model. They found that Wnt-mediated function requires the region of extracellular Cysteine Rich Domain. Experiments in this study were well designed and the data were compelling. The conclusion is supported by the experimental results.

We thank the reviewer for the positive comments on the quality of the work.

Major Concerns:The authors should perform dot blot, solid phase, and/or biolayer interferometry analysis experiments to measure the binding affinity of Wnts binds to APP.

Wnts are notoriously “sticky” proteins in vitro and we were not convinced that such experiments will yield easy to interpret results. Nonetheless, we did attempt solid-phase measurements using commercial ELISA kits. We performed the solid-phase experiment with pre-coated anti-flag ELISA plate (Genescript, Cat.No L00455C). First, we tested if the anti-flag plate work by detecting the APP-flag from cell lysate of HEK 293 transfected with APP-flag or GFP control plasmid and follow the official manual. Briefly, 100ul cell lysate (APP-flag or GFP) of gradient concentration (100%-0.78125%) was added to the wells and incubated for 2 hours at room temperature. After washing, anti-APP antibody was added for another 1hour, then incubated with secondary HRP antibody for 1 hour before adding the TMB substrate. Absorbance was read on a microplate reader at 370 nm. Results, shown in panel A of Author response image 1, indicate the plate works well as the absorbance of APP-flag increased following the increasing concentration but not that of the GFP control.

Then we tried to test the binding affinity between APP-flag and Wnt3a. Different concentrations of Wnt3a (100nM-0.046nM) were added to the wells for 2-3 hours after the incubation of APP-flag, GFP or APP-deltaCRD cell lysate, or PBT to control for binding of Wnt3a alone to the plate. Similarly, we used an “antibody only” test where no Wnt3a was added. This was followed by adding Wnt3a antibody and HRP-conjugated secondary antibody, absorbance was read at 370nm after TMB substrate incubation. As shown in panel B of Author response image 1 the Wnt3a only test shows the same absorbance trend compare with APP, GFP and APP-ΔCRD group. Those results indicate that Wnt3a shows string non-specific binding to the plate, precluding any further testing with this approach. We found exactly the same results with Wnt5a.

**Author response image 1. respfig1:** APP-Wnt binding affinity test by solid-phase.

Significance:This study reveals a new function of APP as a receptor for Wnt and may provide some pieces of evidence to explain why abnormal Wnt pathway regulates Aβ generation and accumulation in AD.

We agree with the reviewer that the physiological function of APP as a Wnt co-receptor is likely to be key to its role in AD and have now new data showing the effects of Wnts on Aβ production in neurons.

Reviewer #3Evidence, reproducibility and clarity:While amyloid-β (Aβ) cleaved from amyloid precursor protein (APP) is a key molecule in the pathogenesis of Alzheimer’s disease (AD), APP and its homologues have been shown to mediate various aspects of neuronal development and activity. In this manuscript, Authors have found that Wnt3a and Wnt5a bind to cysteine rich domain (CRD) of APP and differently regulates APP trafficking and levels, which influence axonal and dendritic growth of neurons. Although their finding is potentially important, they lack the impact because the links between APP and Wnt pathways have been reported. Also, there are several concerns as follow.

We thank the reviewer for the positive comments on the potential importance of the work. We note that both reviewers 1 and 2 described the work as significant and novel. While we agree that a link between APP and Wnt signaling has been described, and our group was one of the first to demonstrate this in vivo, we stress that the exact nature of this link has remained entirely unclear. Showing that APP directly interacts with Wnts ligands and examining why those ligands might have antagonistic effects on APP-dependent processes is a very significant step forward in providing a clear link between the physiological function of APP at the molecular and cellular levels and its potential role in AD. The lack of this link has hitherto been a source of major controversy in the field as to whether APP’s physiological function had any relevance to AD at all. Our data, as well as new experiments we performed provide very strong evidence that this is the case.

Major Concerns:1. Although authors described that APPL mediates Wnt5a function in axonal growth through Figure 1, it is too conclusive because the effect of Wnt5 is not so evident in the figure. The results should be confirmed using Wnt5 overexpression approach in addition to knockout.

We do not agree that Wnt5 overexpression in *Drosophila* is a helpful experiment. The loss of Wnt5 causes little effects on mushroom body β lobe outgrowth, but completely rescues loss Of Vang, but not loss of Appl. These are very strong genetic epistasis data that allow clear conclusions that the effect of Wnt5 on the pathway absolutely requires APP.

2. It is insufficiently addressed how deletion of APP-CRD influence the β-catenin canonical pathway, the PCP signaling pathway and/or the calcium pathway in Wnt signaling.

We and others have previously shown that APP is a key component in both Wnt canonical and non-canonical signaling groups [1,3]. We have now performed IF and WB with β-catenin, Dvll and Dvl2 antibodies. Unfortunately, only the Β-catenin antibody worked in WB experiments on primary neurons. We find that Wnt3a treatment increases levels of β-Catenin in control but not APP-KO primary neurons, consistent with previous findings in cell lines that APP enhances canonical Wnt signaling. These data are now presented in Figure 4—figure supplement 1 and on page 8 of the revised manuscript.

3. Does Wnt induce the endocytosis of Frizzled and LRP5/6 or specific APP? Will be those Wnt receptors co-localized with APP after exposure to Wnt?

While we agree that it is per interesting to know whether Wnts regulate other receptors similarly or differently to APP, we do not feel that this is relevant to our work at this stage. There are many receptors and co-receptors for Wnts, including many members of the Frizzled and LRP families as well DRKs and Vang family members and so on. Studying how each of those responds to Wnt treatment in primary cortical neurons and whether that is correlated to APP dynamics is in and of itself an entire study that requires first establishing whether these receptors are expressed in these neurons, how they are normally localized and then if and how they interact with APP in the presence of absence of APP with or without the CRD domain. This would be very interesting as a future research avenue. We have data showing that in HEK cells Wnt5a addition changes the colocalization of *Drosophila* Vang and APPL in a CRD-dependent manner (Author response image 2) and can add them to the revised manuscript if the reviewer feels this is crucial, but prefer not add this dimension to the current work.

**Author response image 2. respfig2:** Wnt5a regulates APPL-Vang co-localization.

4. To assess APP trafficking, only one time point is not sufficient. It is also unclear how much APP localizes in recycling endosomes after exposure to Wnt?

To address this question, we examined APP distribution in our 3 genetic conditions (APP-WT, APPKO rescued with APP and APPKO rescued APP-ΔCRD) at 2 hours after Wnt addition. We found that Wnt treatment induces re-localization of APP at already at 2 hours with the same pattern observed at 4 hours: Wnt3a caused re-loclaization of APP to the TGN in a CRD-depedent manner while Wnt5a treatment caused re-localization to the lysosome in a CDR-depdent manner. In terms of recycling endosomes, we used Rab11 as a marker (the Rab4 antibody did not work in our hands) We find that neither Wnt3a nor Wnt5a has an effect on the fraction of APP localized to Rab11+ recuclcing endosomes, either at 2 hours or at 4 hours.

These data are now presented in Figure 3—figure supplement 3 and on pages 7, 8, 9, 10 of the revised manuscript.

5. While Wnt may influence APP processing, there is no attempt to assess how deletion of APP-CRD influence APP processing

We are not quite sure what precise modification the reviewer is exactly referring to with the term APP “processing”. We assumed the reviewer is referring to Aβ production. We therefroe tested Aβ40 and Aβ 42 production in primary cortical neurons after 4 hours treatment of Wnt3a or Wnt5a at DIV7. Briefly, after a 4-hour treatment with Wnt3a/5a, supernatant was collected at 4 hours and at 24hours for Aβ detection using MSD kit. Our data shows that a 4-hour Wnt3a/5a treatment has a long-lasting effect on Aβ40/42 production and that those effects are antagonistic. Consistent with Wnt3a favoring recycling of APP we found that Wnt3a treatment significantly decrease Aβ40/42 production. In contrast, Wnt5a treatment, which indices APP internalization into acidic compartments, resulted in an increase in Aβ40/42 producing. Importantly, these data are in accordance with other reports that Wnt/catenin pathway favors non-amyloidogenic APP processing while Wnt/PCP signaling does the opposite.

These data are now presented in Figure 4—figure supplement 2 and on pages 8-9 of the revised manuscript.

To address a role for the CRD in this process, we attempted to restore Fl-APP or APP-ΔCRD expression to APP-KO neurons using viral transduction. This proved to be very difficult as levels of both proteins were significantly lower than what is normally present, especially that of APP-ΔCRD, which was expressed at much lower levels (Author response image 3) precluding normalized quantitative comparison between APP and APP-ΔCRD in terms of Aβ40 and Aβ 42 production. Thus, while it is clear that Wnt treatment alters APP proteolysis and Aβ40 and Aβ 42 production, we can surmise ths is due Wnt binding to the APP CRD but cannot be certain at this stage.

**Author response image 3. respfig3:** APP and APP-ΔCRD protein expression after virus transduction at DIV7.

6. Because different concentrations of Wnt3a and Wnt5a are used at one dosage, it might be difficult to compare their effects properly.

Different concentrations (and even higher than those we use) of Wnt3a and Wnt5a are used in published literature on the topic [2,4–6]. However, we have now tested different doses in the Aβ40/42 experiments and find similar effects for Wnt3a and the same trend for lower doses of Wnt5a.

These data are now presented in Figure 4—figure supplement 2 and on pages 8-9 of the revised manuscript.

7. There is no evidence showing that endogenous levels of Wnt can directly bind to APP-CRD. Binding affinities of Wnt3a and Wnt5a to APP-CRD should be examined.

Wnts are notoriously “sticky” proteins in vitro and we were not convinced that such experiments will yield easy to interpret results. Nonetheless, we did attempt solid-phase measurements using commercial ELISA kits. We performed the solid-phase experiment with pre-coated anti-flag ELISA plate (Genescript, Cat.No L00455C). First, we tested if the anti-flag plate work by detecting the APP-flag from cell lysate of HEK 293 transfected with APP-flag or GFP control plasmid and follow the official manual. Briefly, 100ul cell lysate (APP-flag or GFP) of gradient concentration (100%-0.78125%) was added to the wells and incubated for 2 hours at room temperature. After washing, anti-APP antibody was added for another 1hour, then incubated with secondary HRP antibody for 1 hour before adding the TMB substrate. Absorbance was read on a microplate reader at 370 nm. Results, shown in panel A of Author response image 1, indicate the plate works well as the absorbance of APP-flag increased following the increasing concentration but not that of the GFP control.

Then we tried to test the binding affinity between APP-flag and Wnt3a. Different concentrations of Wnt3a (100nM-0.046nM) were added to the wells for 2-3 hours after the incubation of APP-flag, GFP or APP-deltaCRD cell lysate, or PBT to control for binding of Wnt3a alone to the plate. Similarly, we used an “antibody only” test where no Wnt3a was added. This was followed by adding Wnt3a antibody and HRP-conjugated secondary antibody, absorbance was read at 370nm after TMB substrate incubation. As shown in panel B of Author response image 1 the Wnt3a only test shows the same absorbance trend compare with APP, GFP and APP-ΔCRD group. Those results indicate that Wnt3a shows string non-specific binding to the plate, precluding any further testing with this approach. We found exactly the same results with Wnt5a.

8. Other molecules might interact with APP-CRD in addition to Wnt. Additional experiments are necessary to investigate whether Wnt-APP interaction mediates the phenotypes for neurite outgrowth in Figure 6 and 7.

This is an excellent suggestion to disentangle the effects of each of the two pathways on APP-mediated neurite outgrowth. To address this issue we treated neurons expressing either fl-APP or APP-Δ-CRD with Wnt3a or Wnt5a and measured the effect on APP-dependent axon and dendrite growth on DIV3 and DIV7.

On DIV 3 we found that at DIV3 Wnt3a treatment significantly significant increased total axon length, the length of longest axon and axon branch tips in the presence of fl-APP but not APP-Δ-CRD (revision Figure 2 A, B, C). These data suggest that APP mediates Wnt3a effects on axonal outgrowth. In contrast to Wnt3a, Wnt5a treatment had no effect in the presence of fl-APP but resulted in increased axonal length when the CRD domain was removed. These data show that APP normally protects axons from the effects of Wnt5a at this developmental stage and that the CRD is also required for this. Thus, APP promotes Wnt3a and antagonizes Wnt5a effects on neurons at DIV 3 via the CRD domain.

Similarly, on DIV7 we found a positive Wnt3a effect on axon complexity which was observed in the presence of fl-APP but not APP-Δ-CRD. In contrast Wnt5a had no effect. Finally, we found that treatment with either Wnt3a or Wnt5a decreases the number of the main dendrites in the presence of fl-APP and the effect is absent or even reversed in the presence of APP-Δ-CRD.

Together, these data show that the CRD of APP is required for Wnt3a/5a to regulate neurite outgrowth in cortical primary neurons. We will add these data to the revised manuscript.

These data are now presented in Figure 8 and on pages 11-12 of the revised manuscript.

9. It is more informative to assess how APP overexpression influences Wnt signaling.

We are not in full agreement with this comment. Overexpression is rarely, if ever, more informative than loss of function when trying to determine the physiological function of a protein. There is a very good reason why overexpression models of APP have so far failed to yield profound insight into its physiological activity or to properly model AD.

Significance:It is significant to define physiological roles of APP. It is interesting if Wnt3a and Wnt5a differently mediate Wnt pathway through APP.

We thank the reviewer for this positive comment on the significance of our work.

References

1. Soldano, A., Okray, Z., Janovska, P., Tmejová, K., Reynaud, E., Claeys, A., Yan, J., Atak, Z.K., De Strooper, B., Dura, J.-M., *et al.* (2013). The *Drosophila* Homologue of the Amyloid Precursor Protein Is a Conserved Modulator of Wnt PCP Signaling. PLoS Biol. *11*, e1001562. Available at: https://dx.plos.org/10.1371/journal.pbio.1001562 [Accessed April 5, 2021].

2. Paina, S., Garzotto, D., DeMarchis, S., Marino, M., Moiana, A., Conti, L., Cattaneo, E., Perera, M., Corte, G., Calautti, E., *et al.* (2011). Wnt5a is a transcriptional target of Dlx homeogenes and promotes differentiation of interneuron progenitors in vitro and in vivo. J. Neurosci. *31*, 2675–2687. Available at: www.jneurosci.org [Accessed April 5, 2021].

3. Elliott, C., Rojo, A.I., Ribe, E., Broadstock, M., Xia, W., Morin, P., Semenov, M., Baillie, G., Cuadrado, A., Al-Shawi, R., *et al.* (2018). A role for APP in Wnt signalling links synapse loss with β-amyloid production. Transl. Psychiatry *8*, 179. Available at: http://www.nature.com/articles/s41398-018-0231-6 [Accessed April 1, 2019].

4. Horigane, S.I., Ageta-Ishihara, N., Kamijo, S., Fujii, H., Okamura, M., Kinoshita, M., Takemoto-Kimura, S., and Bito, H. (2016). Facilitation of axon outgrowth via a Wnt5a-CaMKK-CaMKIα pathway during neuronal polarization. Mol. Brain *9*, 8. Available at: https://molecularbrain.biomedcentral.com/articles/10.1186/s13041-016-0189-3 [Accessed April 5, 2021].

5. Clark, C.E.J., Richards, L.J., Stacker, S.A., and Cooper, H.M. (2014). Wnt5a induces Ryk-dependent and -independent effects on callosal axon and dendrite growth. Growth Factors *32*, 11–17. Available at: http://www.tandfonline.com/doi/full/10.3109/08977194.2013.875544 [Accessed April 2, 2021].

6. Li, L., Fothergill, T., Hutchins, B.I., Dent, E.W., and Kalil, K. (2014). Wnt5a evokes cortical axon outgrowth and repulsive guidance by tau mediated reorganization of dynamic microtubules. Dev. Neurobiol. *74*, 797–817. Available at: https://onlinelibrary.wiley.com/doi/10.1002/dneu.22102 [Accessed April 2, 2021].